# ON THE MATRIX FORM OF THE QUATERNION FOURIER TRANSFORM AND QUATERNION CONVOLUTION

## ABSTRACT

We study matrix forms of quaternionic versions of the Fourier Transform and Convolution operations. Quaternions offer a powerful representation unit, however they are related to difficulties in their use that stem foremost from non-commutativity of quaternion multiplication, and due to that $\mu^2 = -1$ posseses infinite solutions in the quaternion domain. Handling of quaternionic matrices is consequently complicated in several aspects (definition of eigenstructure, determinant, etc.). Our research findings clarify the relation of the Quaternion Fourier Transform matrix to the standard (complex) Discrete Fourier Transform matrix, and the extend on which well-known complex-domain theorems extend to quaternions. We focus especially on the relation of Quaternion Fourier Transform matrices to Quaternion Circulant matrices (representing quaternionic convolution), and the eigenstructure of the latter. A proof-of-concept application that makes direct use of our theoretical results is presented, where we present a method to bound the Lipschitz constant of a Quaternionic Convolutional Neural Network.

## 1 INTRODUCTION

Quaternions are four-dimensional objects that can be understood as generalizing the concept of complex numbers. Some of the most prominent applications of quaternions are in the fields of computer graphics (Vince, 2011), robotics (Daniilidis, 1999; Fresk & Nikolakopoulos, 2013) and quantum mechanics (Adler, 1995; Susskind & Friedman, 2014). Perhaps the most well known use case of quaternions involves representing a rotation in three-dimensional space (Kuipers, 1999; Stillwell, 2008; Eater & Sanderson, 2018). This representation is convenient, as for example a composition of rotations corresponds to a multiplication of quaternions, and both actions are non-commutative. Compared to other spatial rotation representations such as Euler angles, quaternions are advantageous in a number of respects (Förstner & Wrobel, 2016).

A relatively more recent and perhaps overlooked use of quaternionic analysis is in digital signal processing and computer vision (Subakan & Vemuri, 2011; Grigoryan & Agaian, 2014; Rosa et al., 2018; Hsu et al., 2019). The motivation for using quaternionic extensions of standard theoretical signal processing tools such as filtering and the Fourier Transform (FT), is that their most typical application on color images involves treating them as three separate monochrome images. As a consequence, cross-channel dynamics are ignored in this approach. The solution is to treat each color value as a single, unified object, and one way to do so is by representing multimodal pixel values as quaternions. The caveat of this approach would be that standard tools, filters and methods would have to be redefined and remodeled, a process that is typically not always straightforward (a fact that partly motivates this work). Recent works move away from this initial motivation and apply quaternionic and other hypercomplex representations on neural networks, on network inputs as well as intermediate, deep layers (Parcollet et al., 2020; Zhang et al., 2021).

In this paper, our focus is on the quaternionic version of the Discrete Fourier transform (DFT) and its relation to quaternion convolution. The Quaternion Fourier Transform (QFT) was originally proposed by Sangwine (1996), and in effect it provides the means to analyze and manipulate the frequency content of multichannel signals. We argue that, while the QFT has been studied by a

number of works and has found uses in practical applications (Ell & Sangwine, 2007; Subakan & Vemuri, 2011; Li et al., 2015; Hitzer, 2016), its potential has yet to be exploited in its fullest.

A matrix representation form for the Quaternion Fourier Transform and Convolution and their properties are studied in this paper. [1] We show that a number of properties of the DFT matrix form also do extend to the quaternionic domain, in most cases appearing as a more complex version of the original (DFT) property. We focus on its relation to Quaternionic Convolution and its circulant form, extend the well-known results for the non-quaternionic case (Jain, 1989), and show that the eigenstructure of Quaternionic Circulant matrices is closely connected to the Quaternion Fourier Matrix form.

In our opinion, the most important contributions of this paper are the following: a) We find that circulant matrices and Fourier matrices are "still" related in terms of their properties also in the quaternionic domain, albeit in a more nuanced manner. The *left* and *right* QFT provide us with different functions of the convolution kernel *left* eigenvalues. b) The literature on the theory and applications of quaternion matrices focuses on *right* eigenvalues, as the *left* spectrum is more difficult or impossible to compute in general. Our focus here is on left eigenvalues [2], and we show how to compute them using QFT. c) We show that sums, multiples and products of quaternionic circulant matrices can be reconstructed using manipulations in the left spectrum. d) In terms of application, we show that our results find direct use, replacing and generalizing uses of standard real/complex operators. We present a proof-of-concept application that makes use of our results. Specifically, we propose a way to a) compute the spectral norm of Quaternionic Convolution and b) set a bound on the Lipschitz constant of a Quaternionic Neural Network. These operations are shown to be carried out magnitudes faster than an implementation that is oblivious to the presented results.

The paper is structured as follows. We proceed with a short discussion on Section 2 concerning difficulties handling quaternion matrices and therefore operator matrix forms, and outline the net benefits from such a process. In Section 3, we examine the matrix form of the QFT and Quaternion Convolution and present our contributions regarding their properties. We showcase the usefulness of the proposed form with an application in Section 4. We conclude the paper with Section 5. Proofs for our propositions and supplemental material are left to the Appendix.

Let us stress that this work is *not* a review over pre-existing works. The main paper presents our novel research findings, while we have moved most of the theoretical prerequisites to the Appendices (except those that we have denoted as "straightforward").

## 2    IS QUATERNIONIC REPRESENTATION WORTH THE HASSLE?

Major difficulties in handling quaternionic matrix forms for the FT or the convolution operator involve the following points: a) *Multiplication non-commutativity*: Quaternion multiplication is non-commutative, a property which is passed on to q4ternion matrix multiplication. b) *Multiple definitions of convolution and QFTs*. There is a left-side, a right-side, or even a two-sided Fourier transform for quaternions, as well as variations stemming from choices of different FT axes. (An analogous picture holds for quaternion convolution, see Appendix for details). c) *Difficulties with a convolution theorem*: A long discussion exists on possible adaptations of the convolution theorem to the quaternion domain (Ell & Sangwine, 2007; Cheng & Kou, 2019; Bahri et al., 2013; Pei et al., 2001; Ell et al., 2014). The (complex domain) convolution theorem cannot be readily applied for *any* version of the QFT and *any* version of Quaternion Convolution (Cheng & Kou, 2019). Quaternion convolution theorems have however been proposed in the literature: Ell & Sangwine (2007) discuss a convolution theorem for the discrete QFT; Bahri et al. (2013) present results for the two-sided continuous 2D QFT. A version with the commutative bicomplex product operator also exists, holding specifically for complex signals transformed w.r.t. to axis $j$ (Ell et al., 2014). d) *More complex eigenstructure*: Quaternion matrices have a significantly more complex eigenstructure than real or complex matrices. Due to non-commutativity of multiplication, left and right eigenvalues are defined, each corresponding to either the problem $Ax = \lambda x$ or $Ax = x\lambda$ respectively. Even

---

[1] Concerning prior work, to our knowledge only Ell et al. 2014, sec.3.1.1.1 refer to a matrix form of the QFT, without proving or discussing any of its properties, its relation to convolution or their implications.

[2] Most hitherto works focus on right eigenvalues; as Aslaksen (2001) notes, "In general, it is difficult to talk about eigenvalues of a quaternionic matrix. Since we work with right vector spaces, we must consider right eigenvalues."

worse, the number of the eigenvalues of a quaternionic matrix is in general infinite (Zhang, 1997). e) *Quaternionic determinants*: The issue of defining a Quaternion determinant is non-trivial; a function acting with the exact same properties as those of the complex determinant, defined in the quaternion domain, cannot exist (Dyson, 1972).

But what can we gain from a matrix form and the properties of the constituent quaternionic matrices, especially when moving to the quaternion domain creates all sorts of difficulties? In a nutshell, the motivation is that vectorial signals such as color images can be treated in a holistic manner, and our analysis opens up the potential to build more powerful models directly in the quaternionic domain. In general, given a signal processing observational model, wherever a Fourier operator of a Convolutional operator is defined, a matrix form allows us to easily adapt the model to the quaternion domain. This bears at least two advantages: a) The model can be easily redefined by replacing operators with their quaternion versions. b) Perhaps more importantly, solution of the model (i.e., find some parameter vector $\theta$ that fits best to observations) involves directly using properties of the Fourier and Circulant matrices (e.g. using results about eigenvalues of a sum of circulant matrices). Properties of the related non-quaternion operations are used in state-of-the-art signal, image processing/vision and learning methods. As a few examples, we mention models in deblurring (Nan et al., 2020) and deconvolution (Hidalgo-Gavira et al., 2019), manipulating convolution layers of neural networks (Sedghi et al., 2018; Singla & Feizi, 2020) or constructing invertible convolutions for normalizing flows (Karami et al., 2019). These properties are well-known for the real and complex domain (Jain, 1989). With the current work, (many of) these results are adapted to the quaternionic domain, allowing construction of more powerful, expressive models.

## 3 MATRIX FORMS: QUATERNION FOURIER MATRIX, QUATERNION CIRCULANT MATRIX AND THEIR CONNECTION

The main theme of this section is exploring whether and to what extend do the well-known properties and relation between circulant matrices, the Fourier matrix, and their eigenstructure generalize to the quaternionic domain. The core of these properties (see for example Jain (1989) for a complete treatise) is summarized in expressing the convolution theorem in matrix form, as:

$$g = Cf \implies Ag = AA^{-1}\Lambda_C Af \implies G = \Lambda_C F \tag{1}$$

where $g, f \in \mathbb{C}^N$, $C \in \mathbb{C}^{N \times N}$ is a circulant matrix, $G, F \in \mathbb{C}^N$ are the Fourier transforms of $g, f$. As the columns of the inverse Fourier matrix $A^{-1}$ are eigenvectors of any circulant matrix, the convolution expressed by the circulant $C$ multiplied by the signal $f$ is easily written in eq. 1 as a point-wise product of the transform of the convolution kernel $\Lambda_C$ (represented as a diagonal matrix) and the transform $F$. These formulae in practice aid in handling circulant matrices, which when used in the context of signal and image processing models represent convolutional filters. The most important of these operations are with respect to compositions of circulant matrices, as well as operations in the frequency domain.

Note that concerning the majority of the propositions and corollaries that will be subsequently presented, we have moved our proofs to the Appendix.

### 3.1 QUATERNIONIC CIRCULANT MATRICES

Let $C \in \mathbb{H}^{N \times N}$ be a quaternionic circulant matrix. This is defined in terms of a quaternionic "kernel" vector, denoted as $k_C \in \mathbb{H}^N$, with quaternionic values $k_C = [c_0 \ \ c_1 \cdots c_{N-1}]^T$. Then, the element at row $i = 0..N-1$ and column $j = 0..N-1$ is equal to $k_C[i-j]_N$, where we take the modulo-N of the index in brackets. Hence, a quaternionic circulant matrix bears the following form:

$$C^T = \begin{pmatrix} c_0 & c_1 & c_2 & \cdots & c_{N-1} \\ c_{N-1} & c_0 & c_1 & \cdots & c_{N-2} \\ \vdots & \vdots & \vdots & \ddots & \vdots \\ c_1 & c_2 & c_3 & \cdots & c_0 \end{pmatrix}. \tag{2}$$

For any quaternionic circulant $C$, from the definition of the quaternionic circulant (eq. 2) and quaternionic convolution, we immediately have:

**Proposition 3.1.** *The product $Cx$ implements the quaternionic circular* left *convolution $k_C \circledast x$:*

$$(k_C \circledast x)[i] = \sum_{n=0}^{N-1} k_C[i-n]_N x[n], \tag{3}$$

*taken for $\forall i \in [1, N]$, where $x \in \mathbb{H}^N$ is the signal to be convolved, and $[\cdot]_N$ denotes $modulo - N$ indexing (Jain, 1989) (i.e., the index "wraps around" with a period equal to $N$).*

**Proposition 3.2.** *Quaternionic circulant matrices can be written as matrix polynomials:*

$$C = c_0 I + c_1 \tilde{P} + c_2 \tilde{P}^2 + \cdots + c_{N-1} \tilde{P}^{N-1}, \tag{4}$$

*where kernel $[c_0 c_1 \cdots c_N] \in \mathbb{H}^N$ and $\tilde{P} \in \mathbb{R}^{N \times N}$ is the real permutation matrix (Strang, 2019) that permutes columns $0 \to 1, 1 \to 2, \cdots, N-1 \to 0$. The inverse is also straightforward: any such matrix polynomial is also quaternionic circulant.*

**Corollary 3.2.1.** *Transpose $C^T$ and conjugate transpose $C^H$ are also quaternionic circulant matrices, with kernels equal to $[c_0 \ c_{N-1} \cdots c_2 \ c_1]^T$ and $[\overline{c_0} \ \overline{c_{N-1}} \cdots \overline{c_2} \ \overline{c_1}]^T$ respectively.*

### 3.2 QUATERNIONIC FOURIER MATRICES

We define a class of matrices as Quaternionic Fourier matrices, shorthanded as $Q_N^\mu$ for some pure unit quaternion $\mu$ (termed the "axis" of the transform) and $N \in \mathbb{N}$, as follows. The element at row $i = 0..N-1$ and column $j = 0..N-1$ is equal to $w_{N\mu}^{i \cdot j}$, where we have used $w_{N\mu} = e^{-\mu 2\pi N^{-1}}$, raised to the power of the product of $i, j$. Hence, we write:

$$Q_N^\mu = \frac{1}{\sqrt{N}} \begin{pmatrix} w_{N\mu}^{0 \cdot 0} & w_{N\mu}^{1 \cdot 0} & \cdots & w_{N\mu}^{(N-1) \cdot 0} \\ w_{N\mu}^{0 \cdot 1} & w_{N\mu}^{1 \cdot 1} & \cdots & w_{N\mu}^{(N-1) \cdot 1} \\ \vdots & \vdots & \ddots & \vdots \\ w_{N\mu}^{0 \cdot (N-1)} & w_{N\mu}^{1 \cdot (N-1)} & \cdots & w_{N\mu}^{(N-1) \cdot (N-1)} \end{pmatrix}. \tag{5}$$

**Proposition 3.3** (General properties)**.** *For any Quaternionic Fourier matrix we have the following straightforward properties:*

*(1) $Q_N^\mu$ is square, Vandermonde and symmetric.*

*(2) $[Q_N^\mu]^H Q_N^\mu = Q_N^\mu [Q_N^\mu]^H = I$, $Q_N^\mu$ is unitary.*

*(3) The product $Q_N^\mu x$, where $x \in \mathbb{H}^N$, equals the* left *QFT $\mathcal{F}_L^\mu \{x\}$:*

$$F_L^\mu[u] = \frac{1}{\sqrt{N}} \sum_{n=0}^{N-1} e^{-\mu 2\pi N^{-1} nu} f[n]. \tag{6}$$

*(4) The product $\overline{Q}_N^\mu x = Q_N^{-\mu} x$, where $x \in \mathbb{H}^N$, equals the* left *inverse QFT $\mathcal{F}_L^{-\mu} \{x\}$ .*

$$F_L^{-\mu}[u] = \frac{1}{\sqrt{N}} \sum_{n=0}^{N-1} e^{+\mu 2\pi N^{-1} nu} f[n]. \tag{7}$$

*(5) $Q_N^i = A_N$ where $A_N$ is the (standard, non-quaternionic) Fourier matrix of size $N$ (Jain, 1989; Strang, 2019). The standard DFT comes as a special case of the QFT, for axis $\mu = i$ and a complex-valued input $x \in \mathbb{C}^N$.*

*(6) $Q_N^{-\mu} = (Q_N^\mu)^{-1} = \overline{Q}_N^\mu = (Q_N^\mu)^H$.*

*(7) $Q_N^\mu Q_N^\mu = \check{P}$, where $\check{P}$ is a permutation matrix that maps column $n$ to $[N - n]_N$.*

(see also the Appendix concerning definitions of the QFT and quaternion conjugacy). All the above can be confirmed by using the matrix form definition of eq. 5.

Consequently, and in contrast to what holds in the complex domain, we have an infinite number of different Fourier matrices for a given signal length $N$, one for each different choice of pure unit axis $\mu$. Proposition 3.3 stated that by flipping the sign of the axis we obtain the inverse QFT with respect to the same axis. In general, two arbitrary Fourier matrices are connected via a rotation of their components:

**Proposition 3.4.** *Let $Q_N^\mu$, $Q_N^\nu$ Quaternionic Fourier matrices with non-collinear axes $\mu, \nu$. We can always find unit $p \in \mathbb{H}$ such that*

$$Q_N^\mu = p Q_N^\nu \bar{p} \tag{8}$$

*The required quaternion $p$ is $e^{\xi\theta/2}$, where $\xi = \nu\mu + V(\nu) \cdot V(\mu)$ and $\theta = arccos(V(\mu) \cdot V(\nu))$.*

Let us add a short comment on the intuition behind this relation. We must note that all elements of $Q_N^\mu$ are situated on the same plane in $\mathbb{H}$, which is different from the plane for elements of $Q_N^\nu$ (due to the assumption of non-collinearity). However, for any given axis, all planes intersect the origin and all pass from $(1, 0, 0, 0) \in \mathbb{H}$, since element $[Q_N^\mu]_{1,1}$ equals 1 for any $\mu$ and $N \in \mathbb{N}$. Elements of $Q_N^\mu$ are situated on a unit circle on their corresponding plane, whereupon their position is defined by their relative angle to the line passing between the origin and $(1, 0, 0, 0)$. This position is determined by the value of the element $w_{N\mu}$, and the power to which this element is raised, so as to give the element of $Q_N^\mu$ in question. Intuitively, this action represents a rotation of the unitary disc where all elements of $Q_N^\mu$ are situated.

### 3.3 CONNECTION OF CIRCULANT AND FOURIER MATRICES

The following results can then be proved, underpinning the relation between Quaternionic Circulant and Quaternionic Fourier matrices, in particular with respect to the left spectrum of the former:

**Proposition 3.5** (Circulant & Fourier Matrices). *For any $C \in \mathbb{H}^{N \times N}$ that is circulant, and any pure unit $\mu \in \mathbb{H}$,*

*(1) Any column $k = 1..N$ of the inverse QFT matrix $Q_N^{-\mu}$ is an eigenvector of $C$. Column $k$ corresponds to the $k^{th}$ component of the vector of left eigenvalues $\boldsymbol{\lambda}^\mu = [\lambda_1^\mu \lambda_2^\mu \cdots \lambda_N^\mu]^T \in \mathbb{H}^N$. Vector $\boldsymbol{\lambda}^\mu$ is equal to the right QFT $\mathcal{F}_{R*}^\mu$ of the kernel of $C$.*

*(2) Any column $k = 1..N$ of the inverse QFT matrix $Q_N^{-\mu}$ is an eigenvector of $C^H$. Column $k$ corresponds to the $k^{th}$ component of the vector of left eigenvalues $\boldsymbol{\kappa}^\mu = [\kappa_1^\mu \kappa_2^\mu \cdots \kappa_N^\mu]^T \in \mathbb{H}^N$. The conjugate of the vector $\boldsymbol{\kappa}^\mu$ is equal to the left QFT $\mathcal{F}_{L*}^\mu$ of the kernel of $C$.*

where transforms denoted with an asterisk (*) refer to using a unitary coefficient instead of $1/\sqrt{N}$:

$$F_{L*}^\mu[u] = \sum_{n=0}^{N-1} e^{-\mu 2\pi N^{-1} nu} f[n], \qquad F_{R*}^\mu[u] = \sum_{n=0}^{N-1} f[n] e^{-\mu 2\pi N^{-1} nu}. \tag{9}$$

Note here that the left spectrum becomes important concerning the circulant matrix eigenstructure. In general, quaternionic matrices have two different spectra, corresponding to left and right eigenvalues. For real matrices, the two spectra will coincide, however the exact way these two sets are connected is largely unknown in the case of true quaternionic matrices (Zhang, 1997; Macías-Virgós et al., 2022).

**Corollary 3.5.1.** *For any pure unit axis $\mu \in \mathbb{H}$, the conjugates of the eigenvalues $\boldsymbol{\lambda}^\mu$ and $\boldsymbol{\kappa}^\mu$ are also left eigenvalues of $C^H$ and $C$ respectively.*

**Corollary 3.5.2.** *For any pure unit axis $\mu \in \mathbb{H}$, the vector of left eigenvalues $\boldsymbol{\lambda}^\mu$ is a flipped version of $\boldsymbol{\lambda}^{-\mu}$, where the DC component and the $(N/2)^{th}$ component (zero-indexed) remain in place.*

**Proposition 3.6.** *A set of $N$ left eigenvalues that correspond to the eigenvectors - columns of $Q_N^{-\mu}$ for some choice of axis $\mu \in \mathbb{H}$, uniquely defines a circulant matrix $C$. The kernel of $C$ is computed by taking the inverse right QFT of the vector of the $N$ left eigenvalues.*

**Corollary 3.6.1.** *Given (a vector, or ordered set of) $N$ left eigenvalues $\boldsymbol{\lambda}^\mu$, we can use the QFT to reconstruct a circulant matrix $C \in \mathbb{H}^{N \times N}$ with $\boldsymbol{\lambda}^\mu \subset \sigma(C)$. The corresponding eigenvectors are the columns of the QFT matrix $Q_N^{-\mu}$. The resulting matrix $C$ is unique, in the sense that it is the only matrix $\in \mathbb{H}^{N \times N}$ with this pair of left eigenvalues and eigenvectors.*

**Corollary 3.6.2.** *Writing proposition 3.5 in a matrix diagonalization form ($A = S\Lambda S^{-1}$), where the columns of $S$ are eigenvectors and $\Lambda$ is a diagonal matrix of* left *eigenvalues, is not possible. This would require us to be able to write $CQ_N^{-\mu} = Q_N^{-\mu}\Lambda^\mu$; however, the right side of this equation computes right eigenvalues, while proposition 3.5 concerns left eigenvalues.*

Consequently, and despite an only partial generalization of the corresponding well-known properties to quaternionic circulant matrices, we must note that quaternion circulant matrices have the rather singular property of being relatively easy to numerically compute (part of its) left spectrum:

**Corollary 3.6.3.** *For any Quaternionic Circulant $C$, any* right-side QFT *$\mathcal{F}_R^\mu$ (i.e., with respect to arbitrary pure unit axis $\mu$) of quaternionic convolution kernel $k_C$ will result to a vector of left eigenvalues for $C$.*

In general no procedure is known to be applicable to a generic (non-circulant) quaternionic matrix (Zhang, 1997; Macías-Virgós et al., 2022). *Only* its right spectrum can be fully calculated using a well-defined numerical procedure (Le Bihan & Sangwine, 2003).

**Proposition 3.7** (Eigenstructure of sums, products and inverse of circulant matrices)**.**

*Let $L, K \in \mathbb{H}^{N \times N}$ circulant matrices, then the following propositions hold. (Analogous results hold for $L, K$ doubly-block circulant).*

*(a) The sums $L + K$ and products $LK, KL$ are also circulant. Any scalar product $pL$ or $Lp$ where $p \in \mathbb{H}$ also results to a circulant matrix.*

*(b) Given some pure unit axis $\mu \in \mathbb{H}$, let $w$ be the $i^{th}$ column of the inverse QFT matrix $Q_N^{-\mu}$, and $\lambda_i^\mu, \kappa_i^\mu$ are left eigenvalues of $L, K$ with respect to the shared eigenvector $w$. Then, $w$ is an eigenvector of. . .*

*i. $L + K$ with left eigenvalue equal to $\lambda_i^\mu + \kappa_i^\mu$.*

*ii. $pL$ with left eigenvalue equal to $p\lambda_i^\mu$.*

*iii. $Lp$ with left eigenvalue equal to $\lambda_i^{p\mu p^{-1}} p$ .*

*iv. $LK$ with left eigenvalue equal to $\lambda_i^{\kappa_i^\mu \mu [\kappa_i^\mu]^{-1}} \kappa_i^\mu$ if $\kappa_i^\mu \neq 0$ and equal to $0$ if $\kappa_i^\mu = 0$.*

*v. Provided $L^{-1}$ exists, $\lambda_i^\mu w(\lambda_i^\mu)^{-1}$ is an eigenvector of $L^{-1}$ with left eigenvalue equal to $(\lambda_i^\mu)^{-1}$.*

## 3.4 QUATERNIONIC DOUBLY-BLOCK CIRCULANT MATRICES

Block-circulant matrices are block matrices that are made up of blocks that are circulant. Doubly-block circulant matrices are block-circulant with blocks that are circulant themselves. These latter are useful in representing 2D convolution (Jain, 1989). Results for circulant matrices in general are valid also for doubly-block circulant matrices, where the role of the Quaternionic Fourier Matrix $Q_N^{-\mu}$ is taken by the Kronecker product $Q_M^{-\mu} \otimes Q_N^{-\mu}$.

**Proposition 3.8** (Doubly Block-Circulant & Fourier Matrices)**.** *Let $D \in \mathbb{H}^{MN \times MN}$ be doubly block-circulant, with $N \times N$ blocks, and each block is sized $M \times M$. Let $\mu \in \mathbb{H}$ be a pure unit quaternion. Then the product $D \cdot vec(x)$, implements the quaternionic circular* left 2D *convolution $k_D \circledast x$. The operator $vec(x)$ represents column-wise vectorization of signal $x \in \mathbb{H}^{M \times N}$.*

$$(k_C \circledast x)[i, j] = \sum_{m=0}^{M-1} \sum_{n=0}^{N-1} k_D[i - m, j - n]_{M,N} x[m, n], \tag{10}$$

**Proposition 3.9.** *Let $R = Q_M^{-\mu} \otimes Q_N^{-\mu}$ and the invertible coordinate mapping $k \leftrightarrow i + Mj$. For any $C \in \mathbb{H}^{MN \times MN}$ that is doubly block-circulant and any pure unit $\mu \in \mathbb{H}$,*

*(1) Any column $k = 1..MN$ of $R$ is an eigenvector of $C$. Column $k$ corresponds to the $(i, j)$ component of the matrix of* left *eigenvalues $\boldsymbol{\lambda}^\mu \in \mathbb{H}^{M \times N}$. Matrix $\boldsymbol{\lambda}^\mu$ is equal to the* right QFT *$\mathcal{F}_{R*}^\mu$ of the kernel of $C$.*

*(2) Any column $k = 1..MN$ of $R$ is an eigenvector of $C^H$. Column $k$ corresponds to the $(i, j)$ component of the matrix of* left *eigenvalues $\boldsymbol{\kappa}^\mu \in \mathbb{H}^{M \times N}$. The conjugate of the matrix $\boldsymbol{\kappa}^\mu$ is equal to the* left QFT *$\mathcal{F}_{L*}^\mu$ of the kernel of $C$.*

# 4 PROOF-OF-CONCEPT APPLICATION: FAST COMPUTATION OF SINGULAR VALUES & BOUNDING OF THE LIPSCHITZ CONSTANT

We present a method to bound the Lipschitz constant of a Neural Network that comprises Quaternionic Convolutions, based on our theoretical results. Bounding the Lipschitz constant has been shown to be a way to control the generalization error (Prince, 2023). In turn, the value of the constant depends on the product of the spectral norms of the weight matrices of the network. Given any linear layer, a bound $c$ on its spectral norm can be set by projecting the operation matrix onto the set of linear transforms with a bounded norm (Lefkimmiatis et al., 2013; Sedghi et al., 2018). We are interested in computing and constraining the maximum value of $||f^\ell(x)||$, where $f^\ell$ represents the operator in layer $l$ and $x$ is the layer input, for all layers in the network. For an arbitrary linearity expressed as matrix $B \in \mathbb{H}^{M \times N}$, this leads to dealing with a Rayleigh quotient of the form $\frac{x^H B^H B x}{x^H x}$, and from here we can use the Quaternion SVD (Sangwine & Le Bihan, 2006) to proceed. We analyze $B$ as $U, \Sigma, V^H = qsvd(B)$ and reconstruct the regularized operation as $U \check{\Sigma} V^H$, where $\check{\Sigma}$ contains singular values projected in the range $[0, c]$. Note that in $\mathbb{H}$, singular values are related to the *right* spectrum (cf. Appendix). When the linearity in question corresponds to a Quaternionic Convolution, matrix $B$ is circulant. While we can still use the QSVD over $B$ to bound the spectrum, the required related space and time complexity is prohibitive in practice (we refer to this method as "brute-force", see further below for a comparison and discussion). We can instead circumvent computing the "expensive" doubly-block circulant matrix altogether, using our results.

We will present the approach as a two-step process. First, we will show how to compute singular values for the convolution operation. Second, we will discuss how we can reconstruct a convolution using the clipped singular values.

*Fast Computation of Singular Values.* We require the additional lemmas:

**Lemma 4.1.** *For any pure unit $\mu \in \mathbb{H}$ and any $N \in \mathbb{N}$, the right span of the columns of $Q_N^\mu$ is $\mathbb{H}^N$. For any pure unit $\mu \in \mathbb{H}$ and any $M, N \in \mathbb{N}$, the right span of the columns of $Q_M^\mu \otimes Q_N^\mu$ is $\mathbb{H}^{MN}$.*

**Lemma 4.2.** *Let $A \in \mathbb{H}^N$ be block-diagonal. The left eigenvalues of each block of $A$ are also left eigenvalues of $A$. The right eigenvalues of each block of $A$ are also right eigenvalues of $A$.*

The first key result for computing singular values is:

**Proposition 4.3.** *Let $C \in \mathbb{H}^{N \times N}$ be quaternionic circulant. Then the square norm $||Cx||^2$, given some $x \in \mathbb{H}^N$, can be written as $c^H \Xi c$, where $c, x \in \mathbb{H}^N$ are related by a bijection, and $\Xi = \xi(k_C)$ is a block-diagonal matrix. All blocks of $\Xi$ are either $1 \times 1$ or $2 \times 2$, and are Hermitian. The $1 \times 1$ blocks are at most two for 1D convolution, and at most four for 2D convolution.*

Matrix $\Xi$ is important, because it contains sufficient information to compute the spectrum of the convolution. The notation $\xi(k_C)$ stresses that we only need the kernel of the convolution to construct $\Xi$. The structure of the $2 \times 2$ blocks of $\Xi$ will be of the form:

$$\begin{bmatrix} |\lambda_m|^2 & \bar{\lambda}_m^\| \lambda_n^\perp + \bar{\lambda}_m^\perp \lambda_n^\| \\ \bar{\lambda}_n^\| \lambda_m^\perp + \bar{\lambda}_n^\perp \lambda_m^\| & |\lambda_n|^2 \end{bmatrix}, \tag{11}$$

where $\|$ and $\perp$ refer to simplex and perplex components w.r.t. axis $\mu$ (cf. Appendix, or e.g. Ell & Sangwine 2007), and $\lambda$ terms are *left* eigenvalues of the convolution *kernel* $k_C$. Its $1 \times 1$ blocks are simply equal to a square-magnitude $|\lambda_m|^2$. Indices $m, n$ must form a pair as described in the Appendix.

**Proposition 4.4.** *All singular values can be computed exactly as either the magnitude of a left eigenvalue of $C$, or the square-root of one of the two eigenvalues of the $2 \times 2$ blocks of $\Xi$, again each dependent on a pair of left eigenvalues of $C$. The maximum value of $||Cx||^2$ under the constraint $||x|| = 1$ is given by the square of the largest right eigenvalue of $\Xi = \xi(k_C)$.*

The above results tell us that we can compute the spectrum of our convolution without building the circulant matrix explicitly. Note that in the non-quaternionic case, $\Xi$ would be diagonal, as off-diagonal terms would vanish, and all $2 \times 2$ blocks would become diagonal. In $\mathbb{H}$, we require the extra step of computing singular values for the $2 \times 2$ blocks.

*Bounding the Lipschitz constant.* In order to set a bound on the spectral norm, we need to be able to move in the opposite direction – from a clipped $\tilde{\Xi}$ matrix, to new left eigenvalues, back towards the new kernel. The following corrolary of the previous propositions is a necessary step:

**Corollary 4.4.1.** *For any matrix $\Phi$ such that $\Phi^H \Phi = \xi(k_C)$, matrices $C$ and $\Phi$ have the same singular values.*

The choice of $\Phi$ is not unique, and ideally we want a matrix that will be convenient in the context of clipping and reconstruction. We choose $\Phi$ so that it is block-diagonal, with block structure identical to that of $\Xi$. We pick the $1 \times 1$ blocks to be equal to $[\lambda_m]$, and $2 \times 2$ blocks equal to $\begin{bmatrix} \lambda_m^{\parallel} & \lambda_n^{\perp} \\ \lambda_m^{\perp} & \lambda_n^{\parallel} \end{bmatrix}$ [3]. It is straightforward to confirm that this choice fulfills the requirement set in Corollary 4.4.1. Importantly, this choice of $\Phi$ is related through a bijection with the left eigenvalues of $k_B$. Another bijection exists between left eigenvalues of $k_B$ (cf. corollary 3.6.1) for a given axis, so exact reconstruction of the required clipped convolution is possible.

By making use of the above results, the spectral norm of a given convolutional layer can be clipped as follows: 1) Compute left eigenvalues of the filter using the right QFT, given a choice of $\mu$ (following the results of proposition 3.3). 2) Compute block-diagonal elements of $\Phi$ (cf. corollary 4.4.1). For $1 \times 1$ blocks we simply copy the appropriate left eigenvalue. For $2 \times 2$ blocks we decompose the two left eigenvalues into simplex and perplex parts, as in eq. 11. 3) Compute QSVD for $\Phi$; this can be implemented as QSVDs for the $2 \times 2$ blocks of $\Phi$. Clip singular values and reconstruct $\tilde{\Phi}$. 4) Reconstruct filters using inverse right QFTs (cf. proposition 3.6). 5) Clip *spatial* range of resulting filters to original range (e.g. Sedghi et al. 2018, Section 3). All convolutions are clipped and reconstructed with this process. The rest of the linear components are clipped and reconstructed as described in the beginning of this Section.

*Results and Discussion: Complexity of Space and Time Requirements.* Direct clipping using QSVD is very expensive both in terms of space and time complexity. The input to QSVD is the doubly-block circulant matrix that corresponds to the convolution filter, and it consists of $N^4$ quaternionic elements; our approach does not require computing the doubly-block circulant matrix at all. In terms of time complexity, direct clipping using QSVD requires $\mathcal{O}(N^6)$, the cost of SVD over $C$. Regarding our method, for the forward and inverse QFT, we need 4 FFTs, as each QFT is computed via 2 standard FFTs (Ell & Sangwine, 2007). Also, we require $\approx N^2/2$ eigenvalue decompositions for the $2 \times 2$ blocks of $\Phi$ (depending on how many $1 \times 1$ blocks exist, which in turn depends on whether $N$ is odd or even). Hence, total complexity is in the order of $\mathcal{O}(N^2 log N)$.

*Computing Singular Values Clipping Time Comparison versus Brute-force QSVD.* We have tested runtime of our method versus the brute-force QSVD approach, over a set of random filters. Results can be examined in Table 1, where we compare runtimes using the method we described, versus the brute-force method (direct clipping). Times to just compute singular values, or perform the full clipping and reconstruction process, are also reported. Concerning the size parameter in a practical

Table 1: Runtime comparison for computation of Quaternionic Convolution Singular Values and Spectral Norm Clipping & Reconstruction. Convolution size refers to $2D$ kernel side size. Average CPU time in milliseconds is reported.

| Convolution size | 4 | 8 | 16 | 32 | 64 | 128 |
|---|---|---|---|---|---|---|
| SVs (ours) | 4 | 16 | 80 | 210 | 708 | 1,795 |
| SVs (brute-force) | 5 | 95 | 1,244 | 69,319 | *hours* | *hours* |
| Full (ours) | 5 | 20 | 113 | 280 | 1,119 | 2,590 |
| Full (brute-force) | 8 | 121 | 1,866 | 150,000 | *hours* | *hours* |

application, note that while small convolutional kernel sizes as much as $3 \times 3$ or $5 \times 5$ are prevalent in both real-valued and hypercomplex network architectures, when we express convolution in terms of a circulant or doubly-block circulant matrix we need to take into account the *zero-padded* version of the kernel. In simple terms, when $Cx$ expresses a convolution over input $x \in \mathbb{H}^N$, circulant $C$ must

---

[3] Another choice can be: $\Phi = \Xi^{1/2}$. However it will allow us to reconstruct left eigenvalues only w.r.t their magnitude.

match the dimension of the input, so $C \in \mathbb{H}^{N \times N}$. Hence, when comparing resource requirements for different sizes of convolution, larger magnitudes for $N$ are more relevant to real-world practice. On operation sizes $N \geq 128$, using the brute-force method is practically impossible due to both space and time constraints. On the contrary, our method scales well on larger sizes. A visualization of a clipped and reconstructed filter can be examined in Appendix D.

*Application on a Neural Network.* We test a ResNet32 architecture on CIFAR10. We replaced standard convolutional layers with depthwise convolutions and pointwise convolutions (Chollet, 2017) on the quaternionic domain. We report the results of training both architecture variants considered (QResNet32-small and QResNet32-large), with and without clipping, in Table 2. We observe that *clipping always boosts final performance, regardless of the architecture considered.*

To further understand the dynamics of the clipping procedure, we present the loss and accuracy curves in Figure 1 for the case of *QResNet32-small*, where the per epoch loss and accuracy are reported for both the with and without clipping variants. As we can see, the clipping impact is mostly realized when the scheduling step occurs (i.e., at 40 epochs), indicating that it enables a finer tuning of the network.

| architecture | w/o clipping | w/ clipping |
|---|---|---|
| QResNet32-small | 82.77 | 84.16 |
| QResNet32-large | 80.73 | 85.29 |

Table 2: Comparison of final test accuracy, when setting a bound on the Lipschitz constant (w/ clipping) versus not (w/o clipping). Figures for two considered architectures are reported. Spectral norms are clipped to value $c = 1.0$.

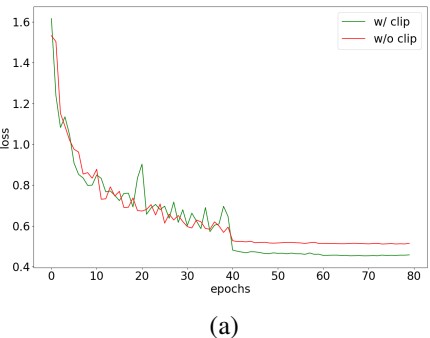

(a)

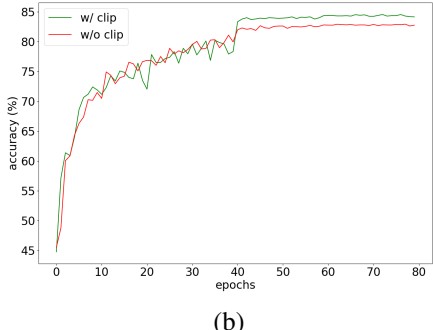

(b)

Figure 1: Per-epoch analysis of the impact of using the proposed technique. Training loss (a) and test accuracy (b) curves are reported.

Please see the Appendix (Section E) for a detailed discussion and ablation over the compared architectures and hyperparameters.

## 5 CONCLUSION

We have presented a set of results concerning the properties and relation between the Quaternion Fourier Transform and Quaternion Convolution, with respect to their formulations in terms of Quaternionic matrices. This relation has been well-known to be very close and important on the real and complex domains, and already used in a wide plethora of learning models and signal processing methods. With our results, we show how and to what extend these properties generalize to the domain of quaternions, opening up the possibility to construct analogous models built directly on quaternionic formulations. We have also presented a method for bounding of the Lipschitz constant using our theoretical results. We plan on working on building competitive quaternionic models, For example, circulant matrices have been used to represent prior terms in various inverse problems, such as image restoration (Chantas et al., 2009) or color image deconvolution (Hidalgo-Gavira et al., 2018). In the derivation of the solution for these models, properties analogous to the ones presented here are a requirement. With the current work, quaternionic versions of these models can in perspective be envisaged.

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

APPENDIX

## A  NOTATION CONVENTIONS

Throughout the text, we have used the following conventions. Capital letters $M, N$ will refer to matrix dimensions, input or filter size. On the "Proof-of-concept" Section 4, $N$ refers to the side size of the input and output images, or equivalently, the zero-padded kernels. Letters $\mu,\nu$ refer to pure unit quaternions, usually used as axes of a QFT. A QFT matrix is written as $Q_N^\mu$, referring to the matrix with side $N \times N$ and reference to axis $\mu$. Letters $\kappa,\lambda$ refer to left eigenvalues. They will be used with an exponent, $\lambda^\mu$, which is to refer to a left eigenvalue with respect to axis $\mu$. When we want to state that a left eigenvalue is related to a specific column of a QFT matrix as its eigenvector, we add a subscript, $\lambda_i^\mu$. Our "by default" ordering is hence with respect to the order of QFT columns (and not e.g. left eigenvalue magnitude). A vector or set of left eigenvalues is denoted boldface, $\boldsymbol{\lambda}^\mu$. A conjugate element is denoted by a bar ($\bar{p}$). An element that has been produced after singular value clipping in the sense discussed in Section 4 is denoted with a "check" character on top: $\check{\Sigma}, \check{\Xi}$.

Indexing, if not stated otherwise, is by default considered to be zero-indexed and "modulo-N" (so index 0 is the same as index $N$, or $-1$ is the same as $N-1$ and so on). This convention is used in order to match the $w_N^\mu$ exponents definition of the QFT (eq. 5), which is very important for most of the subsequent theoretical results.

## B  PRELIMINARIES ON QUATERNION ALGEBRA

In this Section, we shall attempt to outline a selection of important definitions, results and key concepts concerning quaternions (Alfsmann et al., 2007; Cheng & Kou, 2016; Ell & Sangwine, 2007; Ell et al., 2014; Le Bihan, 2017). The structure $\mathbb{H}$ represents the division algebra that is formed by quaternions Quaternions share a position of special importance compared to other algebraic structures, as according to Frobenius' theorem (Fraleigh, 2002), every finite-dimensional (associative) division algebra over $\mathbb{R}$ must be isomorphic either to $\mathbb{R}$, or to $\mathbb{C}$, or to $\mathbb{H}$. Quaternions $q \in \mathbb{H}$ share the following basic form:

$$q = a + b\boldsymbol{i} + c\boldsymbol{j} + d\boldsymbol{k}, \tag{12}$$

where $a, b, c, d \in \mathbb{R}$ and $\boldsymbol{i}, \boldsymbol{j}, \boldsymbol{k}$ are independent imaginary units, Real numbers can be regarded as quaternions with $b, c, d = 0$, and complex numbers can be regarded as quaternions with $c, d = 0$. Quaternions with zero real part, i.e. $a = 0$, are called pure quaternions. For all three imaginary units $\boldsymbol{i}, \boldsymbol{j}, \boldsymbol{k}$, it holds $\boldsymbol{i}^2 = \boldsymbol{j}^2 = \boldsymbol{k}^2 = -1$. The length or magnitude of a quaternion is defined as $|q| = \sqrt{q\bar{q}} = \sqrt{\bar{q}q} = \sqrt{a^2 + b^2 + c^2 + d^2}$, where $\bar{q}$ is the conjugate of $q$, defined as $\bar{q} = a - b\boldsymbol{i} - c\boldsymbol{j} - d\boldsymbol{k}$. Quaternions with $|q| = 1$ are named unit quaternions. For any unit pure quaternion $\mu$, the property $\mu^2 = -1$ holds. Exponentials of quaternions $e^x$ for $x \in \mathbb{H}$ can be defined through their Taylor series:

$$e^x = \sum_{n=0}^{\infty} \frac{x^n}{n!}. \tag{13}$$

From eq. 13, Euler's identity extends for quaternions, for $\mu$ unit and pure:

$$e^{\mu\theta} = cos\theta + \mu \sin\theta. \tag{14}$$

Note also the caveat that in general $e^{\mu+\lambda} \neq e^\mu e^\lambda$. In particular, for $\alpha, \beta \in \mathbb{R}^+$ and two distinct pure unit quaternions $\mu, \nu$, we have $e^{\nu\alpha}e^{\mu\beta} \neq e^{\nu\alpha+\mu\beta}$. Equality holds however, when quaternions $\mu, \nu$ commute. Hence,

$$e^{\mu\alpha}e^{\mu\beta} = e^{\mu\alpha+\mu\beta}, e^{\mu\alpha}e^\beta = e^{\mu\alpha+\beta}. \tag{15}$$

since real numbers commute with any quaternion. Furthermore, any quaternion can be written in polar form as:

$$q = |q|e^{\mu\theta}. \tag{16}$$

Unit pure quaternion $\mu$ and real angle $\theta$ are called the eigenaxis and eigenangle (or simply axis and angle or phase) of the quaternion (Alexiadis & Daras, 2014). The eigenaxis and eigenangle can be computed as: $\mu = V(q)/|V(q)|, \theta = \tan^{-1}(|V(q)|/S(q))$. For pure $q$, hence $S(q) = 0$, we have $\theta = \pi/2$.

In terms of number of independent parameters, or "degrees of freedom" ($DOF$), note that the magnitude $|q|$ corresponds to $1DOF$, $\mu$ to $2DOFs$ (a point on a sphere), and $\theta$ to $1DOF$, summing to a total of $4DOF$. An alternative way to represent quaternions, is by writing their real and collective imaginary part separately. In particular:

$$q = S(q) + V(q), \tag{17}$$

where $S(q) = a$ and $V(q) = b\boldsymbol{i} + c\boldsymbol{j} + d\boldsymbol{k}$. Quaternion multiplication is in general non-commutative, with:

$$\boldsymbol{ij} = -\boldsymbol{ji} = \boldsymbol{k}, \boldsymbol{jk} = -\boldsymbol{kj} = \boldsymbol{i}, \boldsymbol{ki} = -\boldsymbol{ik} = \boldsymbol{j}. \tag{18}$$

Note the analogy of the above formulae to vector products of 3d standard basis vectors. Indeed, these are generalized with the formula for the product of two generic quaternions $p, q \in \mathbb{H}$:

$$\begin{aligned} pq = S(p)S(q) - V(p) \cdot V(q) \\ + S(p)V(q) + S(q)V(p) + V(p) \times V(q), \end{aligned} \tag{19}$$

where $\cdot$ and $\times$ denote the dot and cross product respectively.

*Quaternion matrices, eigenvalues and eigenvectors*: Matrices where their entries are quaternions will be denoted as $A \in \mathbb{H}^{M \times N}$. An important complication of standard matrix calculus comes with matrix eigenstructure and determinants. Due to multiplication non-commutativity, we now have two distinct ways to define eigenvalues and eigenvectors. In particular, solutions to $Ax = \lambda x$ are *left* eigenvalues and eigenvectors, while solutions to $Ax = x\lambda$ are *right* eigenvalues and eigenvectors. The sets of left and right eigenvalues are referred to as left and right spectrum, denoted $\sigma_l(\cdot)$ and $\sigma_r(\cdot)$ respectively. The two sets are in general different, with differing properties as well. Both sets can have infinite members for finite matrices, unlike real and complex matrices. We note here the following important lemmas on the left and right spectrum (Huang & So, 2001): We have

$$\sigma_l(pI + qA) = \{p + qt : t \in \sigma_l(A)\},$$

where $A \in \mathbb{H}^{N \times N}$, $p, q \in \mathbb{H}$, $I$ is the identity matrix in $\mathbb{H}^{N \times N}$. This property does not hold for the right spectrum.

We have

$$\sigma_r(U^H AU) = \sigma_r(A),$$

where $A, U \in \mathbb{H}^{N \times N}$ and $U$ is unitary. Also, $\lambda \in \sigma_r(A) \implies q^{-1}\lambda q \in \sigma_r(A)$ for $q \neq 0$. These properties do not hold for the left spectrum. The interested reader is referred to Zhang (1997) for a treatise on the properties of quaternionic matrices.

*Quaternion Singular Value Decomposition*: Zhang (1997) and Le Bihan & Mars (2004) have been among the first works to discuss the extension of the SVD into the quaternionic domain, dubbed Singular Value Decomposition for Quaternion matrices or simply Quaternion Singular Value Decomposition (QSVD). Sangwine & Le Bihan (2006) have proposed a bidiagonalization-based implementation of QSVD.

The fundamental proposition is that, for any matrix $A \in \mathbb{H}^{M \times N}$, or rank $r$, there exist two quaternion unitary matrices $U$ and $V$ such that

$$A = U\Sigma V^H = U \begin{bmatrix} \Sigma_r & 0 \\ 0 & 0 \end{bmatrix} V^H, \tag{20}$$

where $\Sigma_r$ is a real diagonal matrix and has $r$ strictly positive entries on its diagonal (Le Bihan & Mars, 2004, Proposition 10). The elements of the diagonal are termed singular values. Matrices $U \in \mathbb{H}^{M \times M}$ and $V \in \mathbb{H}^{N \times N}$ are made up of columns that are each left and right eigenvectors of $S$. Note that, perhaps confusingly, the adjectives "left" and "right" are not corresponding to the "left" and "right" spectra of the matrix. *Both* sets of eigenvectors correspond to singular values which are in turn rather more closely related to the right spectrum. We can see this by considering $A^H A$ or $AA^H$, which starting from eq. 20 equal to $V\Sigma^2 V^H$ and $U\Sigma^2 U^H$ respectively. Consequently, we have $A^H AV = V\Sigma^2$ and $AA^H U = U\Sigma^2$, which solve the *right* eigenvalue problem, $\tilde{A}x = x\lambda$.

Hence, the singular values are the non-zero *right* eigenvalues of either $A^H A$ or $AA^H$. As these must be in $\mathbb{R}$, as eigenvalues of Hermitian matrices, they are also *left* eigenvalues. Left eigenvalues that are not right eigenvalues may very well exist, even for Hermitian matrices (Zhang, 1997, Example 5.3).

*Cayley-Dickson form and symplectic decomposition*: A quaternion may be represented as a complex number with complex real and imaginary parts, in a unique manner. We write

$$q = A + B\boldsymbol{j}, \tag{21}$$

with

$$A = q_1 + q_2\boldsymbol{i}, B = q_3 + q_4\boldsymbol{i}.$$

An analogous operation can be performed for quaternion matrices, which can be written as a couple of complex matrices (Zhang, 1997). This scheme can be easily generalized to using any other couple of perpendicular imaginary units $\mu_1, \mu_2$ instead of $\boldsymbol{i}, \boldsymbol{j}$. In that, more general case, it is referred to as a symplectic decomposition (Ell & Sangwine, 2007), which is essentially a change of basis from $(1, \boldsymbol{i}, \boldsymbol{j}, \boldsymbol{k})$ to new imaginary units $(1, \mu_1, \mu_2, \mu_3)$. We can in general write

$$p = p^{\|} + p^{\perp},$$

where $p^{\|} = p_0 + p_1\mu_1$ and $p^{\perp} = (p_2 + p_3\mu_1)\mu_2 = p_2\mu_2 + p_3\mu_3$. Quaternions $p^{\|}$ and $p^{\perp}$ are referred to as the parallel / simplex part and the perpendicular / perplex part with respect to some basis $(1, \mu_1, \mu_2, \mu_3)$. Given symplectic decompositions for quaternions $p, q$, their parts may commute or *conjugate-commute* (Ell & Sangwine, 2007, Section 7C). Note the conjugate operator over $p^{\|}$ in the second relation:

$$p^{\|}q^{\|} = q^{\|}p^{\|}, \qquad p^{\|}q^{\perp} = q^{\perp}\bar{p}^{\|}.$$

## C QUATERNIONIC CONVOLUTION AND QUATERNIONIC FOURIER TRANSFORM

### C.1 QUATERNIONIC CONVOLUTION

In the continuous case, quaternionic convolution is defined as (Bahri et al., 2013):

$$(f * g)(x) = \int_V f(u)g(x - u)du, \tag{22}$$

and cross-correlation (Ell et al., 2014) as:

$$(f \star g)(x) = \int_V \overline{f(u)}g(x + u)du, \tag{23}$$

where $V = \mathbb{R}, \mathbb{R}^2$ for 1D and 2D convolution/correlation respectively. Similarity of the above to the well-known non-quaternionic formulae is evident. A number of useful properties of non-quaternionic convolution also hold for quaternionic convolution. These properties include linearity, shifting, associativity and distributivity (Bahri et al., 2013). Perhaps unsurprisingly, quaternionic convolution is non-commutative ($f * g \neq g * f$ in general), and conjugation inverses the order of convolution operators ($\overline{f * g} = \bar{g} * \bar{f}$). In practice, discrete versions of 1D convolution and correlation are employed. In this work we will use the following definitions of 1D quaternion convolution (Ell et al., 2014, Section 4.1.3):

$$(h_L * f)[n] = \sum_{n=0}^{N-1} h_L[i]f[n - i], \tag{24}$$

$$(f * h_R)[n] = \sum_{n=0}^{N-1} f[n - i]h_R[i], \tag{25}$$

where the difference between the two formulae is whether the convolution kernel elements multiply the signal from the left or right. Extending to 2D quaternion convolution, two of the formulae correspond precisely to left and right 1D convolution:

$$(h_L * f)[n, m] = \sum_{n=0}^{N-1}\sum_{m=0}^{M-1} h_L[i, j]f[n - i, m - j], \tag{26}$$

$$(f * h_R)[n, m] = \sum_{n=0}^{N-1}\sum_{m=0}^{M-1} f[n - i, m - j]h_R[i, j], \tag{27}$$

while a third option is available, in which convolution kernel elements multiply the signal from both left and right (Ell et al., 2014):

$$(h_L \prec f \succ h_R)[n, m] =$$
$$\sum_{n=0}^{N-1} \sum_{m=0}^{M-1} h_L[i, j] f[n - i, m - j] h_R[i, j]. \tag{28}$$

This last variant, referred to as bi-convolution (Ell et al., 2014), has been used to define extension of standard edge detection filters Sobel, Kirsch, Prewitt for color images. A variation of bi-convolution has also been used in Zhu et al. (2018) as part of a quaternion convolution neural network, locking $h_L = \overline{h}_R$ and adding a scaling factor.

We introduce circular variants to the above formulae. Circular left convolution is written as:

$$(h_L * f)[n] = \sum_{n=0}^{N-1} h_L[i]_N f[n - i], \tag{29}$$

where $[\cdot]_N$ denotes $modulo - N$ indexing (Jain, 1989).

## C.2 Quaternionic Fourier Transform

For $1D$ signals we have the definition[4] of the left- and right-side QFT:

$$F_L^\mu[u] = \frac{1}{\sqrt{N}} \sum_{n=0}^{N-1} e^{-\mu 2\pi N^{-1} nu} f[n], \tag{30}$$

$$F_R^\mu[u] = \frac{1}{\sqrt{N}} \sum_{n=0}^{N-1} f[n] e^{-\mu 2\pi N^{-1} nu}, \tag{31}$$

where $\mu$ is an arbitrary pure unit quaternion that is called the *axis* of the transform. For 2D signals, these formulae are generalized to the following definitions (Ell & Sangwine, 2007):

$$F_L^\mu[u, v] = \frac{1}{\sqrt{MN}} \sum_{m=0}^{M-1} \sum_{n=0}^{N-1} e^{-\mu 2\pi (M^{-1} mv + N^{-1} nu)} f[n, m], \tag{32}$$

$$F_R^\mu[u, v] = \frac{1}{\sqrt{MN}} \sum_{m=0}^{M-1} \sum_{n=0}^{N-1} f[n, m] e^{-\mu 2\pi (M^{-1} mv + N^{-1} nu)}. \tag{33}$$

We shall also employ the short-hand hand notation $\mathcal{F}_X^\mu\{g\}$ with $X = \{L, R\}$ to denote the left or right QFT with axis $\mu$ of a signal $g$. The inverse transforms will be denoted as $\mathcal{F}_X^{-\mu}$, as we can easily confirm that $\mathcal{F}_X^\mu \circ \mathcal{F}_X^{-\mu} = \mathcal{F}_X^{-\mu} \circ \mathcal{F}_X^\mu$ is the identity transform. Asymmetric definitions are also useful, and we will use $\mathcal{F}_{X*}^\mu$ to denote a transform with a coefficient equal to 1 instead of $1/\sqrt{N}$ and $1/\sqrt{MN}$.

## C.3 Convolution theorem in $\mathbb{H}$

In the complex domain, the convolution theorem links together the operations of convolution and the Fourier transform in an elegant manner. For most combinations of adaptation variants of convolution and Fourier transform in the quaternionic domain, a similar theorem is not straightforward, if at all possible (e.g. Cheng & Kou 2019). Ell & Sangwine (2007) have proved the following formulae for right-side convolution (we have changed equations to represent n-dimensional signals in general):

$$\mathcal{F}_L^\mu\{f * h\} = \sqrt{N}\{F_{1L}^\mu[u] H_L^\mu[u] + F_{2L}^\mu[u] \mu_2 H_L^{-\mu}[u]\}, \tag{34}$$

$$\mathcal{F}_R^\mu\{f * h\} = \sqrt{N}\{F_R^\mu[u] H_{1R}^\mu[u] + F_R^{-\mu}[u] H_{2R}^\mu[u] \mu_2\}, \tag{35}$$

---

[4]As different transforms are obtained by choosing a different axis $\mu$, we have chosen to denote the axis explicitly in our notation. This formulation is slightly more generic than the one proposed by Ell & Sangwine (2007), and we have the correspondence of $F^{-X}$ (their notation) to $F_X^{-\mu}$ (our notation).

which by symmetry are complemented by the following formulae for left-side convolution:

$$\mathcal{F}_L^\mu\{h * f\} = \sqrt{N}\{H_{1L}^\mu[u,v]F_L^\mu[u,v] + H_{2L}^\mu[u,v]\mu_2 F_L^{-\mu}[u,v]\}, \tag{36}$$

$$\mathcal{F}_R^\mu\{h * f\} = \sqrt{N}\{H_R^\mu[u]F_{1R}^\mu[u] + H_R^{-\mu}[u]F_{2R}^\mu[u]\mu_2\}. \tag{37}$$

On the above formulae, transforms $F_L^\mu, H_R^\mu$ are decomposed as:

$$F_L^\mu = F_{1L}^\mu + F_{2L}^\mu\mu_2, F_R^\mu = F_{1R}^\mu + F_{2R}^\mu\mu_2,$$

$$H_L^\mu = H_{1L}^\mu + H_{2L}^\mu\mu_2, H_R^\mu = H_{1R}^\mu + H_{2R}^\mu\mu_2,$$

where we use symplectic dempositions (cf. section B) with respect to the basis $(1, \mu, \mu_2, \mu\mu_2)$. Note that $\mu_2$ is an arbitrary pure quaternion conforming to $\mu \perp \mu_2$.

## D  RECONSTRUCTION VISUALIZATION OF QUATERNIONIC DOUBLY-BLOCK CIRCULANT MATRIX

In Section 4, we refer to a kernel for our experiments, which was produced using the code that follows. We report the code in the interest of reproducibility of the result. We used the "Quaternion" Python package (Boyle, 2018) (as well as for all implementations required for this paper):

```python
import numpy as np
import quaternion
# Using np.quaternion from https://github.com/moble/quaternion
C_filter = np.zeros([N, N], dtype=np.quaternion)
for i in range(N):
 for j in range(N):
  lm1 = np.cos(i)
  lm2 = np.sin(i)
  lm3 = np.cos(i+.3)
  lm4 = np.sin(i-.2)
  C_filter[i, j] = np.quaternion(i+lm1/N, j+lm2/N, i*j*lm3, i*j*lm4)
```

In Figure 2 we show an example result of clipping and reconstruction using our method (the proof-of-concept method of Section 4 applied over a single filter) versus applying directly the QSVD ("brute-force") on the corresponding doubly-block circulant matrix. The test filter used is a $32 \times 32$ quaternionic convolution. For reference, the mean (maximum) of the singular values of the original kernel is 3005.2 (238519.9) and the clipping threshold was set to 4000. We have used $\mu = (\boldsymbol{i} + \boldsymbol{j} + \boldsymbol{k})/\sqrt{3}$ as the QFT axis, following Ell & Sangwine (2007).

Both singular values and the reconstruction result are exact using our method. The discrepancy of point-to-point difference between our reconstruction and brute-force is in the ballpark of $10^{-11}$ units of magnitude, at worst (cf. last row in Figure 2). In terms of time requirements, our method required 280 milliseconds to obtain an exact reconstruction, compared to $\sim 3$ minutes for the brute-force method.

## E  LIPSCHITZ CONSTANT BOUNDING TESTS ON CIFAR10

We follow Sedghi et al. (2018) and test a ResNet32 architecture. In our model, 3 convolutional blocks, each one consisted of 5 basic residual blocks, are considered, followed by a fully connected layer. Between each block a downsampling operation is used (with a strided convolution). An initial convolutional layer transforms the $3d$ image input into a $16d$ feature tensor.

For our case, we replaced the convolutional layers of basic residual with depthwise convolutions (Chollet, 2017; DeMagistris et al., 2022). Between the blocks, pointwise convolutions are also added to change the number of dimensions and thus each block can safely proceed with depthwise only operations, using the same number of channels. Of course, these depthwise and pointwise operations are replaced with the quaternionic alternatives. Only the first $3 \times 3$ convolution and the last fully connected layer are retained as they are. This model is referred to as *QResNet32-small* and has only $98k$ parameters. An alternative, where between each depthwise convolution of the basic

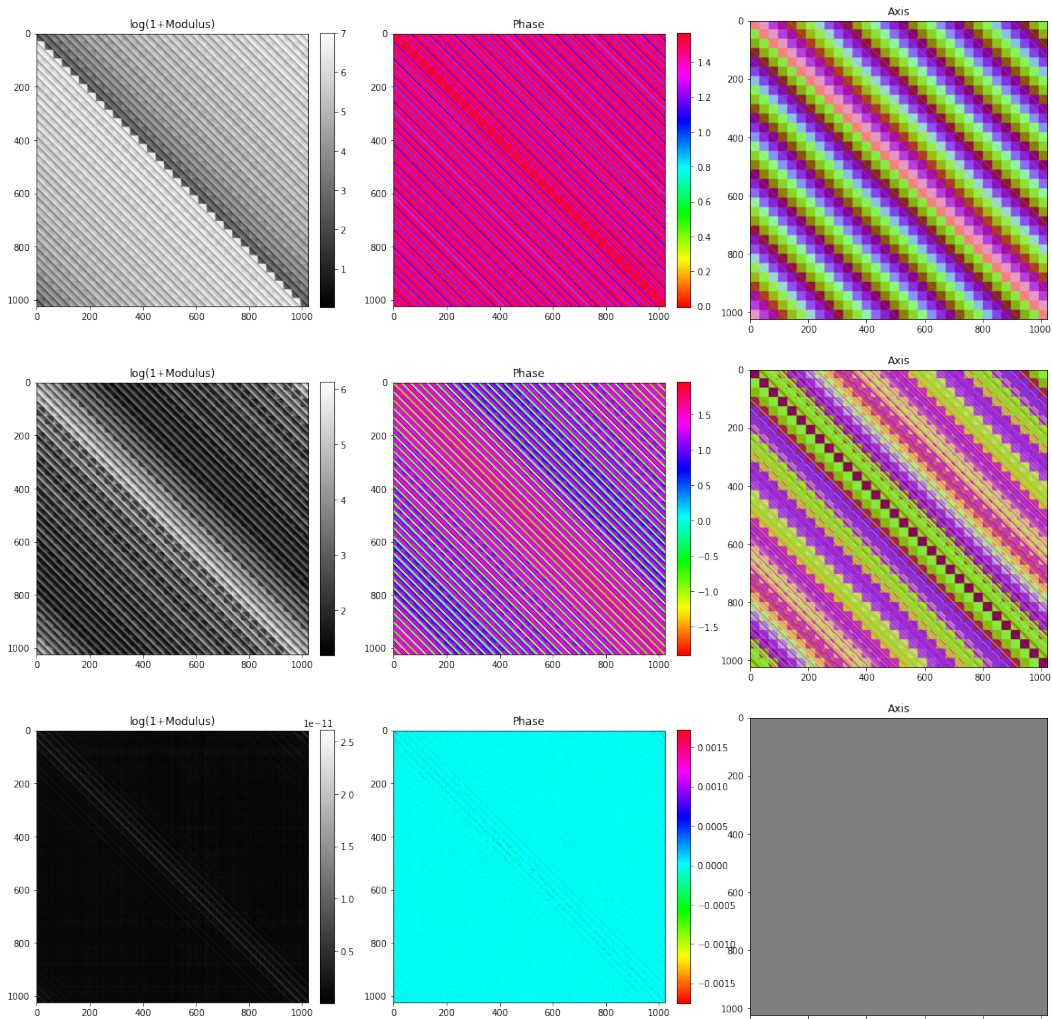

Figure 2: Constraining the spectral norm using QSVD directly on the doubly-block circulant matrix (referred to as "brute-force" in text) vs our method. From top to bottom, we see: the original kernel, the clipped kernel using our method, the point-to-point difference between our result and the brute-force method. From left to right, we see log-magnitudes, phases and axes of respective doubly-block circulant matrices (cf. Section B of the Appendix, or Ell & Sangwine 2007). The brute-force method is considerably more expensive in terms of space as well as time requirements, both by orders of magnitude. For this $32 \times 32$ kernel, our method required 280 milliseconds to obtain an exact reconstruction, while brute-force took over 2 minutes. (see text of Section 4 for more details).

residual block, an extra pointwise convolutional layer is intervened. Its functionality is to assist in the mixing of the channels, at the cost of greatly increasing the number of parameters. Specifically, this version, dubbed as *QResNet32-large* has approximately $454k$ parameters.

Training was performed on CIFAR10, where each model was trained for 80 epochs. Initial learning rate was $0.01$ and was subsequently dropped by $/10$ at 40 and 60 epochs. Clipping is performed every 100 iterations.

We show comparisons in Table 2 and Figure 1 in Section 4 of the main text. Also, to understand how clipping may affect the result, we report the singular values of each quaternionic layer in Figure 3, for training the *QResNet32-small* architecture without clipping. For visualization purposes, each layer is depicted by a mean value and the standard deviation of the corresponding singular values of the layer.

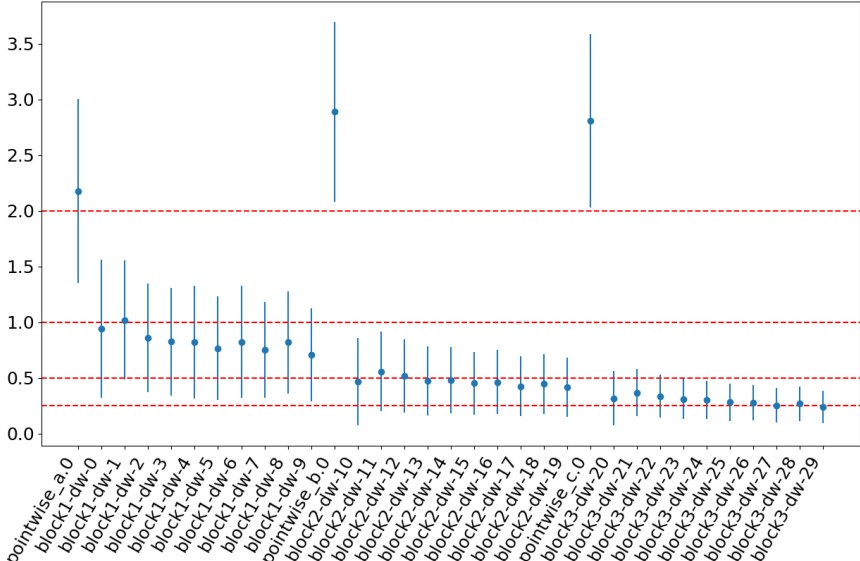

Figure 3: Distribution of the singular values for each layers, when training a *QResNet32-small* architecture. The reported values are for the final model, after 80 epochs.

We can observe the following:

- Clipping always boosts final performance, regardless of the architecture considered.
- The large variant is more sensitive to training without clipping.
- The overall performance of the two architectures is very close. To this end we continue out experimentation with the more "interesting" case of the small version.

Next, we investigate the impact of the clipping value $c$. The results are summarized in Table 3. As we can see, choosing the clipping values affect the overall performance. Specifically, choosing a value from 0.25 to 0.5 seems to be more beneficial, while lower or higher values over-"prune" or are too modest to provide notable impact.

| architecture | $c = 0.10$ | $c = 0.25$ | $c = 0.50$ | $c = 1.00$ | $c = 2.00$ |
|---|---|---|---|---|---|
| QResNet32-small | 83.09 | 85.65 | 85.74 | 84.16 | 82.17 |

Table 3: Impact of the clipping value on the overall performance of the model.

Another aspect that we have test is the impact of the choice of the axis $\mu$ of the QFT. This acts as a hyperparameter, as we need QFT to compute left eigenvalues, and reconstruct the filter from the new left eigenvalues after singular value clipping. In theory, its impact on the performance should be minimal. Towards this, we randomly generated three different $\mu$ axes. Their test accuracy curves are visualized in Figure 4. As we can see, clipping seems to be equally effective, regardless of our choice of the $\mu$ axis.

Note also that the non-quaternionic equivalent, with only depthwise convolutions in the residual blocks, is under-performing, achieving $75.41\%$ test accuracy, even when using more parameters, namely $151K$ parameters in total. We explain this interesting result as follows: Cascades of depthwise convolutions in $\mathbb{R}$ are much less expressive than their counterpart in $\mathbb{H}$, because the former represent series of convolutions over a single real-valued channel. The latter however, do comprise a form of "channel mixing" under the hood, due to the definition of the Hamilton product. Hence,

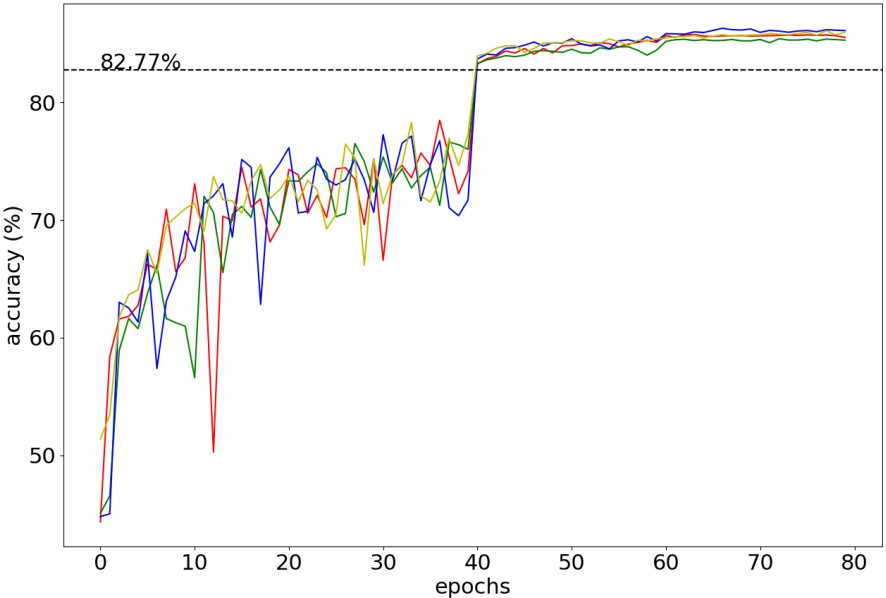

Figure 4: Test accuracy curves during the training for the *QResNet32-small* architecture. Four randomly initialized $\mu$ values are considered for a clipping value of $c = 0.5$. Baseline without clipping is reported as a dashed line at 82.77%.

the quaternionic depthwise layers stand as an interesting trade-off between the completely independent convolutions of real-valued depthwise convolution, and full convolution, taking into account correlation between all possible pairs of channels.

## F  PROOFS FOR PROPOSITIONS AND COROLLARIES IN THE MAIN TEXT

**Corollary 3.2.1** Transpose $C^T$ and conjugate transpose $C^H$ are also quaternionic circulant matrices, with kernels equal to $[c_0 \ c_{N-1} \cdots c_2 \ c_1]^T$ and $[\overline{c_0} \ \overline{c_{N-1}} \cdots \overline{c_2} \ \overline{c_1}]^T$ respectively.

Proof: We use $\tilde{P}^H = \tilde{P}^T = \tilde{P}^{-1}$, and $\tilde{P}^{N-a}\tilde{P}^a = I$ for any $a = 0..N$. We then have, by using Proposition 3.2:

$$C^T = c_0\tilde{P}^N + c_1\tilde{P}^{N-1} + c_2\tilde{P}^{N-2} + \cdots + c_{N-1}\tilde{P}, \tag{38}$$

and

$$C^H = \overline{c_0}\tilde{P}^N + \overline{c_1}\tilde{P}^{N-1} + \overline{c_2}\tilde{P}^{N-2} + \cdots + \overline{c_{N-1}}\tilde{P}, \tag{39}$$

hence both are polynomials over $\tilde{P}$, thus quaternionic circulant. $\square$

**Proposition 3.3** (General properties). (7) $Q_N^\mu Q_N^\mu = \check{P}$, where $\check{P}$ is a permutation matrix that maps column $n$ to $[N-n]_N$.

These properties are a straightforward generalization from $\mathbb{C}$; let us only add a short comment on the derivation of (7) however. We practically need to prove that each column of $Q_N^\mu$ is perpendicular to exactly one other column of the same matrix, and this mapping is given by $n \to N-n$ (where we use a zero/modulo-N convention, cf. Appendix A). For columns $\alpha_u$ and $\alpha_v$, we compute the result of the required product at position $(u, v)$ as:

$$\alpha_v^T \alpha_v = \sum_{n=0}^{N-1} e^{-\mu 2\pi N^{-1}nu} e^{-\mu 2\pi N^{-1}nv} = \sum_{n=0}^{N-1} e^{-\mu 2\pi N^{-1}nu} e^{-\mu 2\pi N^{-1}nv} e^{+\mu 2\pi N^{-1}Nu},$$

which holds since $e^{\mu 2\pi u} = 1$. This is subsequently equal to

$$\sum_{n=0}^{N-1} e^{+\mu 2\pi N^{-1}n(N-u)} e^{-\mu 2\pi N^{-1}nv} = \alpha_{N-u}^H \alpha_v,$$

which is the inner product of columns $v$ and $N-u$. This will result in zero for all pairs of $v, N-u$ except for when $v = [N-u]_N$ (because of e.g. Proposition 3.3(2)). Hence, the required product will have all-zero columns save for exactly one element, different for each column; this is by definition a permutation matrix (Strang, 2019). $\square$

**Proposition 3.4** Let $Q_N^\mu$, $Q_N^\nu$ Quaternionic Fourier matrices with non-collinear axes $\mu, \nu$. We can always find unit $p \in \mathbb{H}$ such that

$$Q_N^\mu = pQ_N^\nu \bar{p}. \tag{40}$$

The required quaternion $p$ is $e^{\xi\theta/2}$, where $\xi = \nu\mu + V(\nu) \cdot V(\mu)$ and $\theta = arccos(V(\mu) \cdot V(\nu))$.

Proof: For any $w_{N\mu}$ and $w_{N\nu}$, we can find $p \in \mathbb{H}$ such that $w_{N\mu} = pw_{N\nu}p^{-1}$. Since any pure unit quaternions $\nu, \mu$ are situated on the unit sphere $\{t \in \mathbb{R}\boldsymbol{i} + \mathbb{R}\boldsymbol{j} + \mathbb{R}\boldsymbol{k} : |t| = 1\}$, we can obtain the one from the other by applying a rotation about the origin. Consequently, there exists $p \in \mathbb{H}$ so that $\mu = p\nu p^{-1}$ (Stillwell, 2008) (note that $w_{N\mu}, w_{N\nu}$ are unit but not necessarily pure). From there, assuming pure unit $\mu$ and real $\theta$ we have the following steps: $e^{\mu\theta} = e^{p\nu p^{-1}\theta} = pp^{-1}cos\theta + p\nu p^{-1}sin\theta = p(cos\theta + \nu sin\theta)p^{-1} = pe^{\nu\theta}p^{-1}$.

It suffices to plug $\theta = 2\pi N^{-1}$ in the previous equation to conclude the required $w_{N\mu} = pw_{N\nu}p^{-1}$. The same transform can be used for all powers of $w_{N\mu}, w_{N\nu}$ to any exponent $\gamma$, and indeed we have $pw_{N\mu}^\gamma p^{-1} = pw_{N\mu}p^{-1}pw_{N\mu}p^{-1}p \cdots w_{N\mu}p^{-1} = w_{N\nu}w_{N\nu} \cdots w_{N\nu} = w_{N\nu}^\gamma$. Since any element of $Q_N^\mu$ can be written as a power $w_{N\mu}^\gamma$ for some natural exponent $\gamma$, we can write $Q_N^\mu = pQ_N^\nu p^{-1}$, which applies the same transform on all elements simultaneously. Intuitively, this action represents a rotation of the unitary disc where all elements of $Q_N^\mu$ are situated.

The axis of rotation is given by the cross product of the two pure unit vectors $\mu, \nu$, as it is by definition perpendicular to the plane that $\mu, \nu$ form. The product $V(\nu) \times V(\mu)$ is equal to $\nu\mu + V(\nu) \cdot V(\mu)$ as $S(\mu) = S(\nu) = 0$, hence we compute $\xi = \nu\mu + V(\nu) \cdot V(\mu)$. Also, $V(\mu) \cdot V(\nu) = |\mu||\nu|cos(\theta) \implies \theta = arccos(V(\mu) \cdot V(\nu))$. $\square$

**Proposition 3.5** (Circulant and Fourier Matrices). For any $C \in \mathbb{H}^{N \times N}$ that is circulant, and any pure unit $\mu \in \mathbb{H}$,

(1) Any column $k = 1..N$ of the inverse QFT matrix $Q_N^{-\mu}$ is an eigenvector of $C$. Column $k$ corresponds to the $k^{th}$ component of the vector of *left* eigenvalues $\boldsymbol{\lambda}^\mu = [\lambda_1^\mu \lambda_2^\mu \cdots \lambda_N^\mu]^T \in \mathbb{H}^N$. Vector $\boldsymbol{\lambda}^\mu$ is equal to the *right* QFT $\mathcal{F}_{R*}^\mu$ of the kernel of $C$.

(2) Any column $k = 1..N$ of the inverse QFT matrix $Q_N^{-\mu}$ is an eigenvector of $C^H$. Column $k$ corresponds to the $k^{th}$ component of the vector of *left* eigenvalues $\boldsymbol{\kappa}^\mu = [\kappa_1^\mu \kappa_2^\mu \cdots \kappa_N^\mu]^T \in \mathbb{H}^N$. The conjugate of the vector $\boldsymbol{\kappa}^\mu$ is equal to the *left* QFT $\mathcal{F}_{L*}^\mu$ of the kernel of $C$.

where transforms denoted with an asterisk (*) refer to using a unitary coefficient instead of $1/\sqrt{N}$:

$$F_{L*}^\mu[u] = \sum_{n=0}^{N-1} e^{-\mu 2\pi N^{-1} nu} f[n], \qquad F_{R*}^\mu[u] = \sum_{n=0}^{N-1} f[n] e^{-\mu 2\pi N^{-1} nu}. \tag{41}$$

Proof: Let $\alpha_k$ the $k^{th}$ column of $Q_N^{-\mu}$. Its product with quaternionic circulant $C$ is computed as:

$$\sqrt{N} C \alpha_k = \begin{pmatrix} \sum_{n=0}^{N-1} h[0-n]_N w_{N\mu}^{-kn} \\ \sum_{n=0}^{N-1} h[1-n]_N w_{N\mu}^{-kn} \\ \sum_{n=0}^{N-1} h[2-n]_N w_{N\mu}^{-kn} \\ \vdots \\ \sum_{n=0}^{N-1} h[N-1-n]_N w_{N\mu}^{-kn} \end{pmatrix}, \tag{42}$$

where $k[i]_N$ is the $i^{th}$/modulo-N element of circulant matrix kernel $k$. The $m^{th}$ element of $C\alpha_k$ is then:

$$\begin{aligned} [C\alpha_k]_m &= \frac{1}{\sqrt{N}} \sum_{n=0}^{N-1} h[m-n]_N w_{N\mu}^{-kn} = \frac{1}{\sqrt{N}} \sum_{l=m}^{m-(N-1)} h[l]_N w_{N\mu}^{-kn} \\ &= \frac{1}{\sqrt{N}} \left[ \sum_{l=m}^{m-(N-1)} h[l]_N w_{N\mu}^{kl} \right] w_{N\mu}^{-km} = \sum_{l=0}^{N-1} h[l]_N w_{N\mu}^{kl} \frac{1}{\sqrt{N}} w_{N\mu}^{-km}, \end{aligned} \tag{43}$$

where we used $l = m - n$ and in the last form of equation 43, the term inside the brackets ($\sum_{l=0}^{N-1} h[l]_N w_{N\mu}^{kl}$) we have the $k^{th}$ element of the right-side QFT $\mathcal{F}_{R*}^\mu$ of $h$. (Note that this is the *asymmetric* version of the QFT, cf. Appendix C). Outside the brackets, $\frac{1}{\sqrt{N}} w_{N\mu}^{-km}$ can be identified as the $m^{th}$ element of $\alpha_k$, shorthanded as $[\alpha_k]_m$. Therefore, $C\alpha_k = \lambda\alpha_k$, so $\alpha_k$ is an eigenvector of $C$; by the previous argument, $\lambda$ is equal to the $k^{th}$ element of the right-side QFT. The second part of the theorem, concerning $C^H$, is dual to the first part. $\square$

**Corrolary 3.5.1.** For any pure unit axis $\mu \in \mathbb{H}$, the conjugates of the eigenvalues $\boldsymbol{\lambda}^\mu$ and $\boldsymbol{\kappa}^\mu$ are also left eigenvalues of $C^H$ and $C$ respectively.

Proof: We will prove the corrolary for $\overline{\boldsymbol{\lambda}^\mu}$ and $C^H$ as the case of $\overline{\boldsymbol{\kappa}^\mu}$ and $C$ is dual to the former one. We have $Cw = \lambda_i^\mu w \implies (C - \lambda_\mu^i I)w = 0$ given non-zero eigenvector $w$, hence the nullspace of $C - \lambda_\mu^i I$ is non-trivial. From this, we have $rank[C - \lambda_i^\mu I] < N \implies rank[(C - \lambda_i^\mu I)]^H < N$ and $rank[C^H - \overline{\lambda_i^\mu}] < N$, where we have used the property $rank(A) = rank(A^H)$ (Zhang, 1997, theorem 7.3). Thus, the nullspace of $C^H - \overline{\lambda_i^\mu} I$ is nontrivial, and $\overline{\lambda_i^\mu}$ is an eigenvalue (Note that this proof does not link these eigenvalues to specific eigenvectors, however). $\square$

**Corollary 3.5.2.** For any pure unit axis $\mu \in \mathbb{H}$, the vector of left eigenvalues $\boldsymbol{\lambda}^\mu$ is a flipped version of $\boldsymbol{\lambda}^{-\mu}$, where the DC component and the $(N/2)^{th}$ component (zero-indexed) remain in place.

Proof. We want to prove that the $i^{th}$ component of $\boldsymbol{\lambda}^\mu$ ($\lambda_i^\mu$) is equal to the $[N-i]_N^{th}$ component of $\boldsymbol{\lambda}^{-\mu}$ ($\lambda_{N-i}^\mu$), (where ordering is w.r.t. modulo-N indexing). But the $\lambda_i^\mu$ is a left eigenvalue that corresponds to the $i^{th}$ column of the QFT matrix with axis $\mu$ as its eigenvector, and also equal to the $i^{th}$ component of the right QFT by Proposition 3.5. It is equal to $\sum_{n=0}^{N-1} f[n] e^{-\mu 2\pi N^{-1} ni}$, while the $[N-i]_N^{th}$ component of $\lambda^{-\mu}$ is equal to $\sum_{n=0}^{N-1} f[n] e^{+\mu 2\pi N^{-1} n(N-i)}$ by definition. Note however

that:

$$\sum_{n=0}^{N-1} f[n]e^{-\mu 2\pi N^{-1}ni} = \sum_{n=0}^{N-1} f[n]e^{-\mu 2\pi N^{-1}ni}e^{+\mu 2\pi N^{-1}Nn} = \sum_{n=0}^{N-1} f[n]e^{+\mu 2\pi N^{-1}n(N-i)},$$

because $e^{\mu 2\pi n} = 1$ for any $n \in \mathbb{Z}$ and pure unit $\mu$. Hence $\lambda_i^\mu = \lambda_{[N-i]_N}^{-\mu}$. The only components for which $i = [N-i]_N$ are $i = 0$ (DC component) and $i = N/2$, hence for these we have $\lambda_i^\mu = \lambda_i^{-\mu}$. $\square$

**Proposition 3.6.** A set of $N$ left eigenvalues that correspond to the eigenvectors - columns of $Q_N^{-\mu}$ for some choice of axis $\mu \in \mathbb{H}$, uniquely defines a circulant matrix $C$. The kernel of $C$ is computed by taking the inverse right QFT of the vector of the $N$ left eigenvalues.

Note that while the analogous proposition is known to hold true for non-quaternionic matrices and the inverse Fourier matrix, it is not completely straightforward to show for quaternionic matrices. For complex matrices it can be proven true by using e.g. the spectral theorem (Strang, 2019), however no such or analogous proposition is known concerning quaternionic matrices and their *left* spectrum.

Proof. We shall proceed by proving the required proposition by contradiction. Let $C \neq D \in \mathbb{H}^{N \times N}$ be circulant matrices; all columns of $Q_N^{-\mu}$ for a choice of axis $\mu$ are eigenvectors of both $C$ and $D$ due to proposition 3.5. We then have:

$$[C\alpha_k]_m = [\sum_{l=0}^{N-1} h[l]_N w_{N\mu}^{kl}]\frac{1}{\sqrt{N}}w_{N\mu}^{-km}, \qquad [D\alpha_k]_m = [\sum_{l=0}^{N-1} g[l]_N w_{N\mu}^{kl}]\frac{1}{\sqrt{N}}w_{N\mu}^{-km},$$

for all $m \in [1, N]$, where $h, g$ are the convolution kernels of $C, D$ respectively, and $\alpha_k$ corresponds to the $k^{th}$ column of $Q_N^{-\mu}$. By the assumption we have set in this proof, we have that left eigenvalues are equal for corresponding eigenvectors for either matrix, hence:

$$\sum_{l=0}^{N-1} h[l]_N w_{N\mu}^{kl} = \sum_{l=0}^{N-1} g[l]_N w_{N\mu}^{kl},$$

or, written in a more compact form,

$$\mathcal{F}_{R*}\{h\} = \mathcal{F}_{R*}\{g\}.$$

However due to the invertibility of the (either left or right-side) QFT, we have $h = g \implies C = D$, which contradicts the assumption. Hence, circulant $C$ must be unique. $\square$

**Proposition 3.7.** (Eigenstructure of sums, products and scalar products of circulant matrices).

Let $L, K \in \mathbb{H}^{N \times N}$ circulant matrices, then the following propositions hold. (Analogous results hold for $L, K$ doubly-block circulant).

(a) The sums $L + K$ and products $LK, KL$ are also circulant. Any scalar product $pL$ or $Lp$ where $p \in \mathbb{H}$ also results to a circulant matrix.

(b) Let $w$ be the $i^{th}$ column of the inverse QFT matrix $Q_N^{-\mu}$, and $\lambda_i^\mu, \kappa_i^\mu$ are left eigenvalues of $L, K$ with respect to the shared eigenvector $w$. Then, $w$ is an eigenvector of. . .

i. $L + K$ with left eigenvalue equal to $\lambda_i^\mu + \kappa_i^\mu$.

ii. $pL$ with left eigenvalue equal to $p\lambda_i^\mu$.

iii. $Lp$ with left eigenvalue equal to $\lambda_i^{p\mu p^{-1}}p$.

iv. $LK$ with left eigenvalue equal to $\lambda_i^{\kappa_i^\mu \mu [\kappa_i^\mu]^{-1}}\kappa_i^\mu$ if $\kappa_i^\mu \neq 0$ and equal to 0 if $\kappa_i^\mu = 0$.

v. Provided $L^{-1}$ exists, $\lambda_i^\mu w(\lambda_i^\mu)^{-1}$ is an eigenvector of $L^{-1}$ with left eigenvalue equal to $(\lambda_i^\mu)^{-1}$.

Proof: (a) For proving $L + K$, $pL$ and $Lp$ are circulant, it suffices to use their matrix polynomial form (cf. Proposition 3.2). Then summation and scalar product are easily shown to be also matrix polynomials w.r.t. the same permutation matrix, hence also circulant.

For the product $LK$, we decompose the two matrices as linear combinations of real matrices, writing

$$LK = (L_e + L_i\boldsymbol{i} + L_j\boldsymbol{j} + L_k\boldsymbol{k})(K_e + K_i\boldsymbol{i} + K_j\boldsymbol{j} + K_k\boldsymbol{k}),$$

where $L_e, L_i, L_j, L_k, K_e, K_i, K_j, K_k \in \mathbb{R}^{N \times N}$ and circulant. Computation of this product results in 16 terms, which are all produced by multiplication of a real circulant with another real circulant, and multiplication by a quaternion imaginary or real unit. All these operations result to circulant matrices (for the proof that the product of real circulant matrices equals a circulant matrix see e.g. Jain 1989), as does the summation of these terms. $KL$ is also circulant by symmetry, however not necessarily equal to $LK$, unlike in the real matrix case.

(b) i. $(L + K)w = Lw + Kw = \lambda_i^\mu w + \kappa_i^\mu w = (\lambda_i^\mu + \kappa_i^\mu)w$. Hence $w$ is an eigenvector of $L + K$ with left eigenvalue equal to $\lambda_i^\mu + \kappa_i^\mu$.

ii. $(pL)w = p(Lw) = (p\lambda_i^\mu)w$. Hence $w$ is an eigenvector of $pL$ with left eigenvalue equal to $p\lambda_i^\mu$.

iii. Let $p_u$ a unit quaternion with $p_u = p/||p||$, assuming $p \neq 0$.

$$(Lp)w = ||p||(Lp_u)w = ||p||Lp_u w p_u^{-1}p_u = ||p||Lzp_u, \tag{44}$$

where $z = p_u w p_u^{-1}$ represents a rotation of $w$ to vector $z$ which is also an eigenvector of $L$; that is because the $k^{th}$ element of $w$ is equal to $e^{-ik\mu 2\pi N^{-1}}$. With a similar manipulation as in Proposition 3.4, $z$ is a column of the inverse QFT matrix $Q_N^{-p_u\mu p_u^{-1}}$, i.e. an inverse QFT matrix with an axis different to the original $\mu$. Thus, due to Proposition 3.5, $z$ is an eigenvector of the circulant $L$, the left eigenvalue of which is $\lambda_i^{p_u\mu p_u^{-1}}$. Continuing eq. 44, we write:

$$||p||Lzp_u = ||p||\lambda_i^{p_u\mu p_u^{-1}}zp_u = ||p||\lambda_i^{p_u\mu p_u^{-1}}p_u w p_u^{-1}p_u \implies$$

$$(Lp)w = (\lambda_i^{p_u\mu p_u^{-1}}p)w, \tag{45}$$

hence $w$ is an eigenvector of $Lp$ with left eigenvalue equal to $\lambda_i^{p_u\mu p_u^{-1}}p$.

iv. $LKw = L\kappa_i^\mu w = \lambda_i^{q_u\mu q_u^{-1}}\kappa_i^\mu w$, where $q = \kappa_i^\mu, q_u = q/||q||$ and we used the right scalar product result of (iii), supposing that $q \neq 0$. Hence $w$ is an eigenvector of $LK$ with left eigenvalue equal to $\lambda_i^{q_u\mu q_u^{-1}}\kappa_i^\mu$. If $q = 0$, we cannot use the result of (iii); the resulting eigenvalue is simply equal to 0, since $LKw = 0$.

v. $Lw = \lambda_i^\mu w \implies Lw(\lambda_i^\mu)^{-1} = \lambda_i^\mu w(\lambda_i^\mu)^{-1} \implies w(\lambda_i^\mu)^{-1} = L^{-1}\lambda_i^\mu w(\lambda_i^\mu)^{-1} \implies L^{-1}\lambda_i^\mu w(\lambda_i^\mu)^{-1} = (\lambda_i^\mu)^{-1}\lambda_i^\mu w(\lambda_i^\mu)^{-1} \implies L^{-1}z = (\lambda_i^\mu)^{-1}z$ where $z = \lambda_i^\mu w(\lambda_i^\mu)^{-1}$.

$\square$

**Proposition 3.9** (Doubly Block-Circulant & Fourier Matrices).

Let $R = Q_M^{-\mu} \otimes Q_N^{-\mu}$ and the invertible coordinate mapping $\varpi : k \leftrightarrow i + Mj$. For any $C \in \mathbb{H}^{MN \times MN}$ that is doubly block-circulant and any pure unit $\mu \in \mathbb{H}$,

(1) Any column $k = 1..MN$ of $R$ is an eigenvector of $C$. Column $k$ corresponds to the $(i, j)$ component of the matrix of *left* eigenvalues $\boldsymbol{\lambda}^\mu \in \mathbb{H}^{M \times N}$. Matrix $\boldsymbol{\lambda}^\mu$ is equal to the *right* QFT $\mathcal{F}_{R*}^\mu$ of the kernel of $C$.

(2) Any column $k = 1..MN$ of $R$ is an eigenvector of $C^H$. Column $k$ corresponds to the $(i, j)$ component of the matrix of *left* eigenvalues $\boldsymbol{\kappa}^\mu \in \mathbb{H}^{M \times N}$. The conjugate of the matrix $\boldsymbol{\kappa}^\mu$ is equal to the *left* QFT $\mathcal{F}_{L*}^\mu$ of the kernel of $C$.

Proof:

The proof for the doubly-block circulant / 2D case is analogous the 1D case. Let $\alpha_k$ the $k^{th}$ column of $R = Q_M^{-\mu} \otimes Q_N^{-\mu}$. The $\ell^{th}$ element of $C\alpha_k$ is:

$$[C\alpha_k]_\ell = \frac{1}{\sqrt{M}} \frac{1}{\sqrt{N}} \sum_{m=0}^{M-1} \sum_{n=0}^{N-1} h[q-m, s-n]_{M,N} w_{M\mu}^{-xm} w_{N\mu}^{-yn}$$

$$[\sum_{p=0}^{M-1} \sum_{r=0}^{N-1} h[p,r]_{M,N} w_{M\mu}^{xp} w_{N\mu}^{yr}] \frac{1}{\sqrt{MN}} w_{M\mu}^{-xq} w_{N\mu}^{-ys},$$

(46)

where we used $\varpi(\ell) = (q,s)$ and $p = q - m$ and $r = s - n$. Note that $w_{M\mu}$ and $w_{N\mu}$ commute between themselves because they share the same axis $\mu$, but they do not commute with the kernel element $h$. $\square$

**Lemma 4.1.** For any pure unit $\mu \in \mathbb{H}$ and any $N \in \mathbb{N}$, the right span of the columns of $Q_N^\mu$ is $\mathbb{H}^N$. For any pure unit $\mu \in \mathbb{H}$ and any $M, N \in \mathbb{N}$, the right span of the columns of $Q_M^\mu \otimes Q_N^\mu$ is $\mathbb{H}^{MN}$.

Proof. Let $c \in \mathbb{H}^N$, and let $x = \sum_{n=1}^N a_k c_k$ (Le Bihan & Mars, 2004, definition 2), where $a_k$ is the $k^{th}$ column of $Q_N^\mu$. In matrix-vector notation, we have $x = Q_N^\mu c$. But $Q_N^\mu$ is invertible for any $\mu, N$ (Proposition 3.3), so $c = Q_N^{-\mu} x$. Thus, for any $x \in \mathbb{H}^N$, we can compute right linear combination coefficients $[c_1 \ c_2 \cdots c_N]^T = c$. The proof for $Q_M^\mu \otimes Q_N^\mu$ is analogous.

(An analogous result holds also for the left span; it suffices to take $c^H = Q_N^{-\mu} x^H$). $\square$

**Lemma 4.2.** Let $A \in \mathbb{H}^N$ be block-diagonal. The left eigenvalues of each block of $A$ are also left eigenvalues of $A$. The right eigenvalues of each block of $A$ are also right eigenvalues of $A$.

Proof. For a block $A_{ab}$ that is situated in the submatrix of $A$ within rows and columns $a : b$, take $[0 \ \cdots \ 0 \ x \ 0 \cdots 0]^T$ as an eigenvector of $A$, where $x$ is an eigenvector of $A_{ab}$. $\square$

**Proposition 4.3.** Let $C \in \mathbb{H}^{N \times N}$ be quaternionic circulant. Then the square norm $||Cx||^2$, given some $x \in \mathbb{H}^N$, can be written as $c^H \Xi c$, where $c, x \in \mathbb{H}^N$ are related by a bijection, and $\Xi = \xi(k_C)$ is a block-diagonal matrix. All blocks of $\Xi$ are either $1 \times 1$ or $2 \times 2$, and are Hermitian. The $1 \times 1$ blocks are at most two for 1D convolution, and at most four for 2D convolution.

Proof. We will treat the case of 1D convolution, and note where the proof changes in a non-trivial manner for 2D. We write $x = \sum_{n=0}^{N-1} a_k c_k$, as in Lemma 4.2. From there we have $Cx = C \sum_{n=1}^N a_k c_k = \sum_{n=1}^N \lambda_k^{-\mu} a_k c_k$. Then:

$$||Cx||^2 = (Cx)^H Cx = c^H Q_N^{-\mu} C^H C Q_N^\mu c = \sum_{m=0}^{N-1} \sum_{n=0}^{N-1} \overline{c_m} a_m^H \overline{\lambda_m^{-\mu}} \lambda_n^{-\mu} a_n c_n. \qquad (47)$$

The caveat is that while we know that $a_m, a_n$ are unit-length pairwise orthogonal, which would have most terms in the above sum to vanish, it appears we can't commute the $\overline{\lambda_m^{-\mu}} \lambda_n^{-\mu}$ quaternion scalar terms out of the way. We can however proceeding by writing $\lambda_n^{-\mu}$ as a sum of a simplex and a perplex part, with respect to axis $-\mu$ (Ell & Sangwine, 2007), i.e. $\lambda_n = \lambda_n^{\parallel} + \lambda_n^{\perp}$ (we have dropped the axis superscript in favor of a more clear notation). Eq. 47 follows up as:

$$= \sum_{n=0}^{N-1} \overline{c_n} a_n^H |\lambda^n|^2 a_n c_n + \sum_{n=0}^{N-1} \sum_{m \neq n} \overline{c_m} a_m^H (\overline{\lambda_m}^{\parallel} + \overline{\lambda_m}^{\perp})(\lambda_n^{\parallel} + \lambda_n^{\perp}) a_n c_n$$

$$= \sum_{n=0}^{N-1} \overline{c_n} |\lambda^n|^2 ||a_n||^2 c_n + \sum_{n=0}^{N-1} \sum_{m \neq n} \overline{c_m} a_m^H (\overline{\lambda_m}^{\parallel} + \overline{\lambda_m}^{\perp})(\lambda_n^{\parallel} + \lambda_n^{\perp}) a_n c_n$$

$$= \sum_{n=0}^{N-1} \overline{c_n} |\lambda^n|^2 c_n + \sum_{n=0}^{N-1} \sum_{m \neq n} \overline{c_m} a_m^H (\overline{\lambda_m}^{\parallel} \lambda_n^{\parallel} + \overline{\lambda_m}^{\perp} \lambda_n^{\perp}) a_n c_n$$

$$+ \sum_{n=0}^{N-1} \sum_{m \neq n} \overline{c_m} a_m^H (\overline{\lambda_m}^{\parallel} \lambda_n^{\perp} + \overline{\lambda_m}^{\perp} \lambda_n^{\parallel}) a_n c_n =$$

$$= \sum_{n=0}^{N-1} |\lambda^n|^2 |c_n|^2 + \sum_{n=0}^{N-1} \sum_{m \neq n} \overline{c_m} (\overline{\lambda_m}^{\|} a_m^H a_n \lambda_n^{\|} + \overline{\lambda_m}^{\perp} a_m^T \overline{a_n} \lambda_n^{\perp}) c_n$$

$$+ \sum_{n=0}^{N-1} \sum_{m \neq n} \overline{c_m} a_m^H (\overline{\lambda_m}^{\|} \lambda_n^{\perp} + \overline{\lambda_m}^{\perp} \lambda_n^{\|}) a_n c_n =$$

$$= \sum_{n=0}^{N-1} |\lambda^n|^2 |c_n|^2 + \sum_{n=0}^{N-1} \sum_{m \neq n} \overline{c_m} (\overline{\lambda_m}^{\|} a_m^H a_n \lambda_n^{\|} + \overline{\lambda_m}^{\perp} [a_n^H a_m]^H \lambda_n^{\perp}) c_n$$

$$+ \sum_{n=0}^{N-1} \sum_{m \neq n} \overline{c_m} (\overline{\lambda_m}^{\|} [a_n^T a_m]^H \lambda_n^{\perp} + \overline{\lambda_m}^{\perp} a_m^T a_n \lambda_n^{\|}) c_n. \tag{48}$$

In the above form, the simplex parts commute with eigenvectors $a_n, a_m$, while the perplex parts conjugate-commute with the same eigenvectors. The reason is that all terms of an eigenvector - column of a Fourier matrix - share the same axis $\mu$. For $n \neq m$, we have $a_m \perp a_n$ because any QFT matrix is unitary (Proposition 3.3), and $a_m, a_n$ are different columns of a QFT matrix by assumption. Hence the second sum in eq. 48 vanishes.

The third sum is of special interest, because for each $n \in [0, N-1]$ there will be *at most* one index $m = n'$ over which the term will not vanish. That is because $a_n^T a_m$ is the inner product between $\overline{a_n}$ and $a_m$ (i.e. where the first term is conjugated), so in effect it is a product between a QFT column and another column of the *inverse* QFT (cf. Proposition 3.3). If and only if indices are $n$ and $m = [N - n]_N$, we have $a_n^T a_m = 1$. In other cases, the term vanishes. Consequently, we have:

$$||Cx||^2 = \sum_{n=0}^{N-1} |\lambda_n|^2 |c_n|^2 + \overline{c_m} \{ \overline{\lambda_m}^{\|} \lambda_n^{\perp} + \overline{\lambda_m}^{\perp} \lambda_n^{\|} \} c_n, \tag{49}$$

where we use $m = [N - n]_N$. Also, recall that all left eigenvalues $\lambda_n$ are w.r.t. axis $-\mu$.

Note that, if and only if $n = 0$ or $n = N/2$ the second sum will also vanish; these are the columns of the QFT matrix that contain only real-valued terms. For a doubly-block circulant matrix these will be at most *four* by construction. To see this, we can use Proposition 3.3.7, which in effect says that the inner product of the $m^{th}$ and $n^{th}$ column of $Q$ will equal to 1 only for $m = n = 0$ or $N/2$. If the number of elements $N$ is an odd number, only the case $m = n = 0$ is possible. In either case, this coincides with the number of non-zero elements in the diagonal of the permutation matrix $\check{P} = QQ$.

For 2D convolution and a doubly-block circulant matrix of size $N^2 \times N^2$ [5], we obtain an analogous result. In this case we are interested in the product $(Q_N \otimes Q_N)(Q_N \otimes Q_N)$. This equals to $Q_N Q_N \otimes Q_N Q_N = \check{P} \otimes \check{P}$. The result $\check{P} \otimes \check{P}$ is another permutation matrix, which will have at most 4 non-zero elements in the diagonal, the exact number of which will depend to whether $N$ is odd or even. For the second term in eq. 49 we need $m = \varpi([N - i_n, N - j_n]_{N,N})$ with $(i_n, j_n) = \varpi^{-1}(n)$ instead of $m = [N - n]_N$ of the 1D case, so again we have at most one non-vanishing second term in eq. 49 for each $n \in [0, N-1]$.

We can proceed from eq. 49 by rewriting it in the form $||Cx||^2 = c^H \Xi' c$, where $\Xi' \in \mathbb{H}^{N \times N}$ is Hermitian. For example, for $N = 6$ we have:

$$\Xi' = \begin{bmatrix} |\lambda_0|^2 & 0 & 0 & 0 & 0 & 0 \\ 0 & |\lambda_1|^2 & 0 & 0 & 0 & \overline{\lambda_1}^{\|} \lambda_5^{\perp} + \overline{\lambda_1}^{\perp} \lambda_5^{\|} \\ 0 & 0 & |\lambda_2|^2 & 0 & \overline{\lambda_2}^{\|} \lambda_4^{\perp} + \overline{\lambda_2}^{\perp} \lambda_4^{\|} & 0 \\ 0 & 0 & 0 & |\lambda_3|^2 & 0 & 0 \\ 0 & 0 & \overline{\lambda_4}^{\|} \lambda_2^{\perp} + \overline{\lambda_4}^{\perp} \lambda_2^{\|} & 0 & |\lambda_4|^2 & 0 \\ 0 & \overline{\lambda_5}^{\|} \lambda_1^{\perp} + \overline{\lambda_5}^{\perp} \lambda_1^{\|} & 0 & 0 & 0 & |\lambda_5|^2 \end{bmatrix},$$

---

[5] This is trivially extensible to non-square-shaped filters, i.e. $MN \times MN$.

which, by rearranging indices of coefficients, and corresponding rows and columns, we have $\Xi'$ which is also Hermitian. For the previous example for $\Xi'$, we would have:

$$
\Xi = \begin{bmatrix}
|\lambda_0|^2 & 0 & 0 & 0 & 0 & 0 \\
0 & |\lambda_3|^2 & 0 & 0 & 0 & 0 \\
0 & 0 & |\lambda_1|^2 & \overline{\lambda_1}^{\parallel}\lambda_5^{\perp} + \overline{\lambda_1}^{\perp}\lambda_5^{\parallel} & 0 & 0 \\
0 & 0 & \overline{\lambda_5}^{\parallel}\lambda_1^{\perp} + \overline{\lambda_5}^{\perp}\lambda_1^{\parallel} & |\lambda_5|^2 & 0 & 0 \\
0 & 0 & 0 & 0 & |\lambda_4|^2 & \overline{\lambda_4}^{\parallel}\lambda_2^{\perp} + \overline{\lambda_4}^{\perp}\lambda_2^{\parallel} \\
0 & 0 & 0 & 0 & \overline{\lambda_2}^{\parallel}\lambda_4^{\perp} + \overline{\lambda_2}^{\perp}\lambda_4^{\parallel} & |\lambda_2|^2
\end{bmatrix}.
$$

Furthermore, $\Xi$ is also block-diagonal, and all of its blocks are of size $2 \times 2$, with the exception of at most two (four) $1 \times 1$ blocks for 1D (2D) convolution. We can then write the required rearranging of coefficients as a multiplication by a permutation matrix $P$ (not unique, and in general $\neq \check{P}, \tilde{P}$). Hence, $||Cx||^2 = c'^H P^T \Xi' P c' = c^H \Xi c$, where we have switched places for $c, c'$ for clarity of notation. $\square$

**Proposition 4.4.** All singular values can be computed exactly as either the magnitude of a left eigenvalue of $C$, or the square-root of one of the two eigenvalues of the $2 \times 2$ blocks of $\Xi$, again each dependent on a pair of left eigenvalues of $C$. The maximum value of $||Cx||^2$ under the constraint $||x|| = 1$ is given by the square of the largest *right* eigenvalue of $\Xi = \xi(k_C)$.

Proof. The required maximum is equal to the maximum of the Rayleigh quotient $\frac{x^H C^H C x}{x^H x}$, which obtains its maximum for the maximum right eigenvalue (Macías-Virgós et al., 2022, Proposition 3.3-3.4). Now, since $c, x$ are related through a bijection, maximizing the Rayleigh quotient of $C^H C$ is equivalent to maximizing the Rayleigh quotient of $\Xi$.

As $\Xi$ is Hermitian, we know that all of its right eigenvalues are real-valued (and are also left eigenvalues). Due to it being block-diagonal, its right eigenvalues will be the union of the eigenvalues of its blocks, due to Lemma 4.3. As all blocks are either $1 \times 1$ or $2 \times 2$ by Proposition 4.1, the sought right eigenvalue will either be: a) equal to the value of the $1 \times 1$ block, so equal to $|\lambda_n|^2$, where $\lambda_n$ is the left eigenvalue of $C$ that corresponds to the $n^{th}$ column of QFT matrix with axis $\mu$, or b) equal to the right eigenvalues of the $2 \times 2$ Hermitian matrix:

$$
\begin{bmatrix}
|\lambda_m|^2 & \overline{\lambda_m}^{\parallel}\lambda_n^{\perp} + \overline{\lambda_m}^{\perp}\lambda_n^{\parallel} \\
\overline{\lambda_n}^{\parallel}\lambda_m^{\perp} + \overline{\lambda_n}^{\perp}\lambda_m^{\parallel} & |\lambda_n|^2
\end{bmatrix}. \tag{50}
$$

The right eigenvalues for the latter case are real-valued and can in principle be computed through a complex adjoint mapping and the spectral theorem (Sfikas et al., 2020). (Recall that by $|\lambda_n|$ we denote the left eigenvalue of $C$, produced by taking the $n^{th}$ element of the right QFT with respect to axis *minus* $\mu$ (cf. Proposition 3.3, Lemma 4.3). $\square$

**Corollary 4.4.1.** For any matrix $\Phi$ such that $\Phi^H \Phi = \xi(k_C)$, matrices $C$ and $\Phi$ have the same singular values.

Proof. We can write the change of basis in eq. 47 as $Q_N^{-\mu} C^H C Q_N^{\mu}$, and after we combine it with a rearraging of rows and columns encoded as a left-right multiplication by permutation matrix $P$, we have

$$
\Xi = P^T Q_N^{-\mu} C^H C Q_N^{\mu} P = (Q_N^{\mu} P)^{-1}(C^H C) Q_N^{\mu} P, \tag{51}
$$

so $\Xi$ and $C^H C$ are similar matrices, which entails that they have the same right eigenvalues (Zhang, 1997, Section 7). The same holds for $\Phi^H \Phi$ and $C^H C$, by our assumption that $\Phi^H \Phi = \xi(k_C)$. The singular values of $\Phi$ are the (non-zero) right eigenvalues of $\Phi^H \Phi$, and the singular values of $C$ are the (non-zero) right eigenvalues of $C^H C$, which proves the corollary. $\square$

