# OpenReview forum: "On the Matrix Form of the Quaternion Fourier Transform and Quaternion Convolution"
_ICLR.cc/2024/Conference — Submitted to ICLR 2024_

### Official Review · Reviewer_JYL4 · 2023-10-25

**Soundness:** 4 excellent
**Presentation:** 3 good
**Contribution:** 1 poor
**Rating:** 5
**Confidence:** 3

**Summary:**

The paper studies the quaternion discrete Fourier transform. The authors establish various facts about quaternion valued circulant matrices, their connections with the regular (complex valued) DFT matrices. They use their results to estimate spectral bounds on a quaternion-valued convolution operator that has been employed in neural networks. They show that their method outperforms, in terms of computation time, a more brute-force approach based on the quaternion SVD.

**Strengths:**

The paper establishes facts about quaternionic Fourier transforms, circulant matrices, which may potentially be useful in applications related to the ICLR community. The authors mention several applications, concerning RGB images, orientation estimation, neural networks employing quaternionic convolution layers. The authors share results on one experiment involving such a neural network. They show that their results can be employed to find a necessary upper bound more efficiently than existing methods.

**Weaknesses:**

Overall, the paper reads like a collection of facts looking for an application. The authors do mention connections with several applications, but these connections are not developed in the paper properly to convince a reader that the results would be directly useful.
The experiment performed by the authors is rather a detail in a larger application and does not really motivate a potential reader to delve into the main content. It would have been interesting to see a treatment of an RGB image using a quaternion representation and employing the statements provided in the paper, or similarly an orientation estimation problem whose treatment is facilitated by the results of the paper.

All of this does not mean that the content is wrong or not at all useful in any application, but ICLR may not be right venue for this content. If the authors would like to revise the paper, I strongly suggest including stronger connections to the applications they touch in Section 2.

**Questions:**

The following are minor questions, and I think the main issue is lack of convincing applications for the statements in the main body.
- Second paragraph : While quaternions help associate $i$, $j$, $k$ to each RGB channel, what does, for instance $i \cdot j = k$ in the RGB context? How would we treat an image with more channels, or a color image  represented via hue-saturation-value using quaternions? The point I'm trying to make is that I don't see a real connection between a color image and quaternions beyond a superficial coincidence of the number of channels with imaginary units. I'd be interested to hear if I'm missing something here.

- I would welcome a more detailed development of the quaternionic DFT for orientation sequences -- specifically, does the fact that orientation quaternions have unit norm have any significance in this context?

---

> ### Author Response · Authors · 2023-11-16
> **“ICLR may not be the right venue for this content.”**
>
> *“All of this does not mean that the content is wrong or not at all useful in any application, but ICLR may not be right venue for this content. [..] If the authors would like to revise the paper, I strongly suggest including stronger connections to the applications they touch in Section 2”*
>
> We respectfully disagree. *We believe that ICLR is a very relevant venue for papers where the main point of the work is presenting new theoretical results*.
>
> Here are some examples of previous works where the paper is largely “theoretical”, i.e. the main novelty comes in the form of theoretical results:
>
> * Sedghi, Gupta, Long, “The singular values of convolutional layers”, ICLR 2019
> * Singla, Feizi, “Fantastic Four: Differentiable and Efficient Bounds on Singular Values of Convolution Layers”, ICLR 2021
> The first paper (Sedghi et al.) in fact uses an experimental setup that is based around testing a ResNet on CIFAR. Our experimental setup has been inspired by this work. The second paper (Singla and Feizi) includes tests on MNIST and CIFAR. According to the meta-reviewer, “the paper should be accepted purely on the basis of its theoretical contribution, which enhances our understanding of this important topic” (https://openreview.net/forum?id=JCRblSgs34Z&noteId=WUrNb_MUat1). Both great papers, in our opinion.
>
> Another example of a great paper from ICLR 2023 – Note that there are no experiments:
> * Haase, Hertrich, Loho, “Lower Bounds on the Depth of Integral ReLU Neural Networks via Lattice Polytopes”, ICLR 2023
>
> Another example of a great paper where the focus is not experimental results, published in ICML this time:
>
> * Cohen and Welling, “Group Equivariant Convolutional Networks”, ICML 2023
>
> The list of course is far from being exhaustive.

---

> ### Author Response · Authors · 2023-11-16
> **“the main issue is lack of convincing applications for the statements in the main body”**
>
> *“The experiment performed by the authors is rather a detail in a larger application and does not really motivate a potential reader to delve into the main content.” [..] “the main issue is lack of convincing applications for the statements in the main body.”*
>
> **The application of Section 4 makes use of almost all of our theoretical results**:
>
> * Propositions 3.1 and 3.2 are about left-side quaternion convolution being expressed as a circulant matrix. The bulk of the NN layers uses left-side quaternion convolution. These will be treated as products of the form $Cx$, where $C$ is circulant, and use the subsequent results for circulant matrices.
> * Proposition 3.3 is about the matrix form of the QFT, and 3.4 is about a geometric intuition over this form.  Proposition 3.3 is used in conjunction with..
> * ..Proposition 3.5, where *left eigenvalues* of C are shown to be computable via a right QFT. The corresponding eigenvectors are vectors of the QFT matrix we discussed in 3.3, 3.4. **This computation is at the heart of the application in Section 4 - none of the computations would be possible without it**.
> * Propositions 3.8 and 3.9 are about doubly-block circulant matrices – again, this is absolutely a requirement as we are dealing with 2D inputs (CIFAR images) in the experiments.
> * All the theoretical results in Section 4 are there specifically to motivate and enable the Lipschitz bounding application. We show how we can clip singular values of quaternionic convolution, by the use of an auxiliary “Ξ” matrix.
>
> Note again that all of the lemmas, propositions, corollaries in Sections 3 and 4 are completely novel content.

---

> ### Author Response · Authors · 2023-11-16
> **“reads like a collection of facts looking for an application”**
>
> *“Overall, the paper reads like a collection of facts looking for an application. “*
>
> **This is largely a theoretical paper.** Nevertheless, we feel that the Lipschitz constant bounding application is important in its own right.
>
> **If by the term “collection of facts” it is insinuated that our results are of lesser value, or are “incremental”, please allow us to argue in favor of the opposite view**. We bring up a number of examples, which we believe that are characteristic:
>
> * Proposition 3.5 is about the **eigenstructure of Quaternionic Circulant matrices**, and their connection to the QFT matrix. The connection between Circulants and the DFT matrix is well-known for R or C – in H (set of quaternions), eigenstructure is however necessarily a far more nuanced issue. For a start, we have two different spectra: a left spectrum and a right spectrum, corresponding to left and right eigenvalues respectively. *This is a feature unique to H*. The two spectra are in general different from the other, and no connection is known to hold between them, in terms of, say, computing the left spectrum if one knows about the right spectrum, or vice versa. The literature has mostly explored the right spectrum, for which the have been proposed numerical computation algorithms. For the left spectrum however, works are far more scarce. Proposition 3.5 tells us that Quaternionic Circulant matrices and the QFT are related through the left spectrum of the former, however there has been no hitherto algorithm to compute left eigenvalues (apart from toy examples of 2x2 or 3x3 matrices). *In quaternion matrix algebra, a field which dates as far back as the first half of the $20^{th}$ century, to our knowledge this is the first work where a computational scheme to compute the left spectrum is proposed* (aside the noted small-matrix cases, cf. So, “Quaternionic Left Eigenvalue Problem”, 2005; Marcias-Virgos, “Rayleigh quotient and left eigenvalues of quaternionic matrices”, Linear & Multilinear Algebra 2023). Furthermore, **it is shown to be applied to a real, relevant learning problem**.
> * In the paper, we write that the convolution theorem for R or C can be written (and proved) easily in matrix form. However, this is not possible in H (Corollary 3.6.2). In particular,  “Writing proposition 3.5 in a matrix diagonalization form (A = SΛS−1), where the columns of S are eigenvectors and Λ is a diagonal matrix of left eigenvalues, is not possible. This would require us to be able to write $CQ = QΛ$; however, the right side of this equation computes right eigenvalues, while proposition 3.5 concerns left eigenvalues.” **So, this is another example that looks “easy” in R or C, but becomes non-trivial and more nuanced / complicated in H**.
> * In Proposition 3.7iv, we state (and prove) that if $\lambda$ and $\kappa$ are eigenvalues of circulant matrices L and K, then their *product* LK has the eigenvalue: $\lambda^\nu \kappa^\mu$, where $\nu = \kappa \mu \kappa^{-1}$. This is a result that has no analogue in R or C, and only makes sense in H. Furthermore it generalizes smoothly into H: In a commutative algebra we would have $\kappa$ and $\kappa^{-1}$ cancel out, which doesn’t happen for quaternions. We believe that this also a non-obvious result a priori: A first guess could be that $\lambda^\mu \kappa^\mu$ is the eigenvalue, which turns out to be wrong.
> In proposition 4.3 and corollary 4.4.1, the matrix $\Xi$ is introduced, in order to enable computation and bounding of the required spectra. Note the form:	$\begin{bmatrix}
>     	\lambda_m^{\parallel} & \lambda_n^{\perp} \\\\
>     	\lambda_m^{\perp} 	& \lambda_n^{\parallel} \\\\
> 	\end{bmatrix}$.
> **There is no analogue in R or C, as simplex and perplex components make sense only in H. Furthermore, note that this elegantly generalizes from R/C**: In these domains, the off-diagonal (perplex) components would be zero, and computation would be trivial.

---

> ### Author Response · Authors · 2023-11-16
> **“the results [are not] directly useful” / Impact and relevance to ICLR**
>
> *”The authors do mention connections with several applications, but these connections are not developed in the paper properly to convince a reader that the results would be directly useful.”*
>
> Concerning our mentioning several applications, we do believe that the scope of application of our results is extensive. This work is about two concepts – Fourier / spectral analysis and Convolution – that are at the very heart of a range of fields, from signal processing to pattern recognition and learning systems and representations. Quaternion convolution in particular, is an operation that is prevalent in the (vast?) majority of works on Quaternion Neural Networks (let me reference here the excellent survey of Parcollet, Morchid and Linarès, “A survey of quaternion neural networks”, AI Review 2020). The close relation between Fourier and Convolution has long been known for the real domain, and has been *extensively* used. In a nutshell, this paper shows how this relation generalizes to the quaternionic domain, and presents a proof-of-concept application of our theoretical results.
>
> Any model and method that includes convolution or spectral analysis (in H) will be relevant to the current paper. Convolutions and Fourier are part of learning systems and representations that range from probabilistic, hierarchical models to deep neural nets. Hence, we believe that the “range of impact” of our work is extensive.
>
> We have taken a conscious decision to include this specific application – it combines Quaternions in NNs, bounding the Lipschitz constant, and analyzing singular values of convolutional layers. The latter two have been the subject of interest for the ICLR / NIPS / ICML etc. community. For example:
>
> * Sedghi, Gupta, Long, “The singular values of convolutional layers”, ICLR 2019
> * Singla, Feizi, “Fantastic Four: Differentiable and Efficient Bounds on Singular Values of Convolution Layers”, ICLR 2021
>
> On another note, the proof-of-concept application itself is useful in its own right – it can be seen as a plug-and-play improvement over practically *any* quaternionic CNN .

---

> ### Author Response · Authors · 2023-11-16
> **Intuition regarding mapping of RGB**
>
> “Second paragraph : While quaternions help associate i,j,k, to each RGB channel, what does, for instance, i j = k mean in the RGB context? How would we treat an image with more channels, or a color image represented via hue-saturation-value using quaternions? The point I'm trying to make is that I don't see a real connection between a color image and quaternions beyond a superficial coincidence of the number of channels with imaginary units. I'd be interested to hear if I'm missing something here.”
>
> In the context of representing color images via elements of H, the main advantage is that we are allowed to treat them in a “holistic manner”.
>
> By “holistic manner”, we mean that a multi-dimensional input may be treated as a single entity. For example, assuming that we have an image containing multiple channels, we need to write it as a structure of the form M x N x C . Strictly speaking, this is a *tensor*. Quaternions (and hypercomplex algebra in general) allow us to write this as a matrix.
>
> This may sound like a superficial difference, but it does bear advantages: The field of matrix methods is much more mature than tensor methods. Convolutions and Fourier in the form of matrices (Circulant, Fourier matrices) is one such example, and what this paper does is showing how a range of well-known related results generalize for quaternions. If we chose to stick with the tensor representation, we’d certainly be much more limited in scope. In Section 4, we discuss singular values of quaternion convolutional layers – this would be (currently?) impossible to do the exact same thing with tensors (there is no SVD for 3-way tensors). Cf. the excellent references on this topic:
> * G. Strang, “Linear Algebra and Learning from data”, 2019 (esp. chapter I.12)
> * C.Hillar and LH. Lim, “Most tensor problems are NP-hard”, J.ACM, 2013
>
> Concerning “what does i j = k mean in the RGB context”: This is an interesting question.
>
> Ideally, we’d like to have an algebra where both actions – summation and multiplication – have some direct connection with an intuitive interpretation. We’re ok about summation, supposing that we use an additive color model, but multiplication in the sense of the Hamilton product is indeed difficult to interpret. Our take is that it’d be super interesting if we could correlate multiplication with another intuitive, explainable action, *and* at the same time be able to construct a valid algebraic structure. There is currently work that is ongoing in other hypercomplex structures, e.g. Ruhe, Brandstetter, Forré, “Clifford Group Equivariant Neural Networks”, NeurIPS 2023 to cite one example. Concerning the HSV color space, a connection between the QFT transform axis and the concept of hue is drawn in Ell and Sangwine 2007, “Hypercomplex Fourier Transforms of color images”, IEEE Transactions in Image Processing.
>
> However, non-straightforward interpretability is not necessarily a serious disadvantage. In our “proof-of-concept” application of Section 4, we use convolutional quaternion NNs to do image classification. Therein, the input layer of these NNs contains exactly what you refer to: a mapping of channels to imaginary components of a quaternion matrix. The first layer of the NN maps this image into a series of filtered-out results using the learned quaternion convolutions. (Quaternion convolutions use the Hamilton product in their definition, cf. App B,C). After the cascade of layers we get our class estimate (which btw is improved vs the vanilla QNN, using the proposed Lipschitz bounding..).
>
> The point is that, much like what happens in real-valued NNs, usability / usefulness and interpretability don’t always go together. Although, yes, interpretability would be great to have (explainable AI, we’re looking at you :)

---

> ### Author Response · Authors · 2023-11-21
> **Concerning orientations, quaternions, and the QFT (part 1)**
>
> *"I would welcome a more detailed development of the quaternionic DFT for orientation sequences -- specifically, does the fact that orientation quaternions have unit norm have any significance in this context?"*
>
> Thank you for the comment.
>
> A series of unit-norm quaternions that represent a series of rotations / orientations will translate into the (QFT) quaternion frequency domain as a set of quaternionic components that will be bounded with respect to their magnitude.
>
> In particular, the magnitude of all components will be at most equal to $\sqrt{N}$, where $N$ is the length of the signal.
>
> We can see this by taking the definition of the QFT. Starting by equation (3) in the paper, we write:
>
> $$
>             |F_{L}^\mu[u]| = |\frac{1}{\sqrt{N}}\sum_{n=0}^{N-1}
>             e^{-\mu 2 \pi N^{-1} nu} f[n]| \leq \frac{1}{\sqrt{N}} \sum_{n=0}^{N-1}|e^{-\mu 2 \pi N^{-1} nu}| |f[n]|,
> $$
>
> where both components in the summation are bounded by unity (the exponential term by definition, and the $f[n]$ term by assumption that we're dealing with rotations, hence unit-norm quaternions).
>
> Thus,
>
> $$
>             \frac{1}{\sqrt{N}} \sum_{n=0}^{N-1}|e^{-\mu 2 \pi N^{-1} nu}| |f[n]| = \frac{N}{\sqrt{N}} = \sqrt{N},
> $$
>
> which proves the assertion.

---

> ### Author Response · Authors · 2023-11-21
> **Concerning orientations, quaternions, and the QFT (part 2)**
>
> We have created a toy example of a series of $N=100$ unit-norm quaternions.
> We can visualize these as rotations, in terms of Euler angles (We have used the conversion convention of the "numpy-quaternion" package).
> We plot the signal as a series of the three Euler angles, represented each with a different color:
> https://freeimage.host/i/JnZcpe4
>
> Next, we can use the QFT to suppress high-frequency components. We compute the QFT and zero all components except those around a window of size $[DC-2, DC+2]$ around the DC component.
> The result is a quite smoothed-out orientation series: https://freeimage.host/i/JnZWtRe
>
> Note that this corresponds to one, very specific, way to filter the input series, and supposing that are goal were to put some smoothness constraint here, one could argue that there are indeed many more ways to do it. The simplest idea would be to change the window size, or multiply each frequency component by some magnitude and so on, much like what we would do if we were using the complex DFT. In the Quaternionic FT, we have a much more expressive tool, as all rotation variates are handled "holistically", their correlations can in principle be very much taken into account. Two ways to see this is that a) filter weights are quaternionic, while a DFT handling of each rotation axis would correspond to weights being in the real domain, and b) we have an extra hyperparameter which is the axis $\mu$ of the transform. (The complex DFT can be said to have a fixed axis equal to $i$).
>
> By changing the QFT axis, we'll get different results, for the same low-freq window. We have used $\frac{i+j+k}{\sqrt{3}}$ in the prior example. If we use $\frac{i+j}{\sqrt{2}}$ we'll have: https://freeimage.host/i/JnZvSON
>
> Or with $\mu = k$: https://freeimage.host/i/JnZSJjI . *Note in this example that the Euler angles "wrap around"*, as they are constrained to lie in a "circular" domain in terms of degrees. (we did filtering in the quaternion domain, so this isn't a problem, aside from the visualization per se. Another small "bonus" for using quaternions; for an interesting rant on quaternions vs euler angles see [here](https://github.com/moble/quaternion/wiki/Euler-angles-are-horrible)).
>
> Another result, this time with a random QFT axis: https://freeimage.host/i/JntHLrb .
>
> **The connection of all the above with our paper is this: All linear filtering in the (quaternionic) Frequency domain can be cast in terms of a quaternion convolution; in this work, we prove results that enhance understanding of the relation between convolution and the Fourier transform. We explore the eigenstructure of the related matrix forms -- circulant and QFT matrix. This is important especially in a learning paradigm (as we show in Section 4 of the paper)**. In the particular, the aforementioned filtering example could correspond to a convolution, which is almost invariably used in quaternion neural networks. All aspects of this filter could be cast in terms of a neural network layer (including the QFT axis, and so on). This is equally true for a series of orientations / members of $SO(3)$.

---

### Official Review · Reviewer_ZfTZ · 2023-10-31

**Soundness:** 1 poor
**Presentation:** 1 poor
**Contribution:** 1 poor
**Rating:** 1
**Confidence:** 4

**Summary:**

The paper presents a set of propositions that show the connection between convolution and Fourier transform for quaternions and provide a bound on the spectrum of quaternion matrices via the SVD. The experimental results compare schemes for computation of singular values of quaternion matrices.

**Strengths:**

+ Neural networks / linear algebra with quaternion representations may offer advantages in some applications.

**Weaknesses:**

- The paper presents propositions, but not a single proof. The appendix largely states existing results from literature, in which the connection between the quaternion convolution and FFT, as well as the SVD are already established.
 - No experimental comparison is made for an application problem, the experiments compare the proposed algorithms to naive baselines for the linear algebra problem (SVD truncation). I do not consider norm/eigenvalue estimation to be an application problem, it is also particular to the choice of using QFT.
 - The presentation of the paper also fails to make clear the challenges or benefits of quaternions throughout, in particular,
   * Quaternion representations are not introduced in the main body mathematically, and the difficulties of quaternion convolution are only discussed by reference to prior work.
   * In contrast, the main body and the appendices largely restate standard algebraic derivations for DFT matrix properties and convolution. Analogous properties to the complex case are derived for quaternions with no motivation for why they are relevant to the contributions of the paper or nontrivial.
 - DFT/FFT correspondence for quaternion convolution has already been established in prior work, the paper seems to provide limited novelty, beyond establishing eigenvalue bounds (which are not motivated).


Overall, the paper seems to contain very incremental results for quaternion linear algebra that follow from existing work. The experimental comparisons do not consider alternative methods or previously existing works. The main theoretical results and applications considered in the paper are basic properties of quaternion convolution and DFT and basic linear algebra. I do not see specific relevance of the work to machine learning and I had difficulty gaining insight about the benefits or challenges of using quaternion convolution/DFT from both the body and the appendices of the paper.

EDIT: I expanded and modified my review as I had previously failed to notice the proofs in Appendix F due to the page break on page 20.

**Questions:**

None

---

> ### Author Response · Authors · 2023-11-15
> **“The paper presents propositions, but not a single proof.”**
>
> *“The paper presents propositions, but not a single proof.” [..] “Overall, the paper seems to contain very incremental results for quaternion linear algebra that follow from existing work, and lacks proofs.”*
>
> **There are a total of 7 pages of proofs**. These are found in Appendix F, pages 21-27.
>
> Every single non-trivial statement in the paper is proved and discussed therein - 13 lemmas and propositions in total, with additional proofs for the corollaries. *All proofs in the paper are novel content.*
>
> If you feel that we should include an additional pointer in the text regarding the position of the proofs, or perhaps move the proofs to an earlier appendix, we’d be happy to do so.

---

> > ### Comment · Reviewer_ZfTZ · 2023-11-21
> >
> > Sorry, I missed Appendix F on my prior read-through. I had expected the location of proofs to be stated along with the theorems. I updated my review. I still have many concerns about the paper, especially the focus of the presentation.

---

> ### Author Response · Authors · 2023-11-15
> **“The appendix largely states existing results from literature [..]"**
>
> *“The appendix largely states existing results from literature, in which the connection between the quaternion convolution and FFT, as well as the SVD are already established.”*
>
> Appendix B is titled “Preliminaries” and Appendix C is titled “Quaternionic Convolution and Quaternionic Fourier Transform”. These parts of the text indeed go through results from the literature.
>
> Appendix F contains proofs for all statements (proofs, lemmas, corollaries) that appear in the main text. **These are novel in their entirety.**
>
> If you think that any one of our propositions has already appeared in previous work, we’d definitely like to know. To our knowledge, all topics over which we present novel propositions have not been previously explored (matrix forms of the convolution, qft, their eigenstructure, doubly-block quaternionic circulant matrices, Lipschitz constant for QCNNs, and so on).

---

> ### Author Response · Authors · 2023-11-15
> **“No experimental comparison is made for an application problem”**
>
> *“No experimental comparison is made for an application problem”, [..]  “The experimental comparisons do not consider alternative methods or previously existing works.” [..] “the experiments compare the proposed algorithms to naive baselines for the linear algebra problem (SVD truncation).”*
>
> **There is experimental comparison** (outlined with results in Fig.1, Tables 1 & 2, Appendix D, Fig.2,3,4 and Table 2, ...), **and there is an application problem**. The whole of Section 4 discusses an application, and there is additional content in the Appendices – Appendix E contains additional results and details, Appendix F contains proofs, part of which are specific to the Lipschitz constant bounding application.

---

### Official Review · Reviewer_YJPK · 2023-11-01

**Soundness:** 3 good
**Presentation:** 3 good
**Contribution:** 2 fair
**Rating:** 5
**Confidence:** 3

**Summary:**

The paper presents a study of the Fourier transform and the convolution operation in the quaternion domain. The authors argue that, despite the appealing properties of quaternions for modelling rotations (and other representations), there are some points of quaternions that are _problematic_ (see Sec 2). Thus, they aim to draw connections between the quaternion and standard Fourier matrix and convolutions.

**Strengths:**

Though I am not an expert on the current SOTA on the use of quaternions for ML or signal processing, I identify conceptual value in the paper in Section 3, where, through a series of Propositions and Corollaries, the paper delivers its contribution. A key point in applied terms is the computation of singular values and computing the Lipschitz constant in a particular example.

**Weaknesses:**

Despite its conceptual contribution, as a non expert on quaternion-valued architectures or techniques I found it difficult to identify the impact of this work to the ICLR community.

- First, the paper outlines the _major difficulties_ of quaternions. However, I don't see why non-commutativity, two-sided convolution, versions of a convolution theorem, definition of determinant and the fact that $\mu = -1$ has infinite solutions are problems. If these issues are to be avoided, then why not go back to the real domain? Isn't the desired structure of quaternions what precisely results in these _difficulties_?

- Second, it is not clear how the machinery developed in Sec 3 impacts the machine/representation learning communities. I acknowledge the representation power of quaternions, however, this paper does not exploit this representation in the context of learning. I feel that this work lacks a clearer connection between the stated contributions and the learning task, possibly by showing examples where this representation makes a difference wrt standard (non-quaternion) methods. I also acknowledge the application in Sec 4, however, the body of results presented in Sec 3 is far more general than that particular application.

- There are a few typos, for instance: the _extend_ on which ...

- Lastly, what do the authors mean by _[treating] signals in a holistic manner_?

**Questions:**

Please refer to the previous part

---

> ### Author Response · Authors · 2023-11-15
>
> *“there are some points of quaternions that are problematic (see Sec 2). Thus, they aim to draw connections between the quaternion and standard Fourier matrix and convolutions.” [..] Despite its conceptual contribution, as a non expert on quaternion-valued architectures or techniques I found it difficult to identify the impact of this work to the ICLR community. [..] Second, it is not clear how the machinery developed in Sec 3 impacts the machine/representation learning communities.*
>
> Thank you for your comments. We will definitely use the feedback to clarify parts of the text that have apparently been misleading.
> Let us clarify that in Section 2 we merely provide a short discussion on the pros and cons of using quaternions as part of a learning representation framework. Perhaps it has been misunderstood that the point of the paper is arguing that “quaternions are worth using”. We have included Section 2 as a short intro for scholars that have perhaps heard that quaternions are useful for e.g. representing rotations, but do not know about the *modern* uses of quaternions in learning systems.
>
> We take for granted that quaternions form a useful framework in ML and closely connected fields. **The paper is not about whether quaternions are useful or not**; we believe that this has been extensively argued in previous literature. Let us reference a few characteristic works:
>
> * Parcollet, Ravanelli, Morchid, Linarès, Trabelsi, De Mori and Yoshua Bengio: “Quaternion Recurrent Neural Networks”, ICLR 2019
> * Zhang, Tay, Zhang, Chan, Luu, Hui and Fu: “Beyond fully-connected layers with quaternions: Parameterization of hypercomplex multiplications with 1/n parameters”, ICLR 2021
> * Zhang, Tao, Yao, Liu, “Quaternion knowledge graph embeddings”, NeurIPS 2019
> * Miron, Flamant, Le Bihan, Chainais and Brie, “Quaternions in Signal and Image Processing”, IEEE Signal Processing Magazine 2023
> * Qin, Zhang, Xu, Xu, “Fast Quaternion Product Units for Learning Disentangled Representations in SO3”, IEEE Transactions on Pattern Analysis and Machine Intelligence, 2023
>
> The first and second works have been published in ICLR. The second one has got an “outstanding paper award” in ICLR. Hence, quaternions in the context of learning representations have indeed interested the community, and have been very much welcomed by it in the recent past.
>
> **Concerning the impact of this work**: This work is about two concepts – Fourier / spectral analysis and Convolution – that are at the very heart of a range of fields, from signal processing to pattern recognition and learning systems and representations. Quaternion convolution in particular, is an operation that is prevalent in the (vast?) majority of works on Quaternion Neural Networks (let me reference here the excellent survey of Parcollet, Morchid and Linarès, “A survey of quaternion neural networks”, AI Review 2020). The close relation between Fourier and Convolution has long been known for the real domain, and has been *extensively* used. In a nutshell, this paper shows how this relation generalizes to the quaternionic domain, and presents a proof-of-concept application of our theoretical results.
>
> We have taken a conscious decision to include this specific application – it combines Quaternions in NNs, bounding the Lipschitz constant, and analyzing singular values of convolutional layers. The latter two have been the subject of interest for the ICLR / NIPS / ICML etc. community. For example:
>
> * Sedghi, Gupta, Long, “The singular values of convolutional layers”, ICLR 2019
> * Singla, Feizi, “Fantastic Four: Differentiable and Efficient Bounds on Singular Values of Convolution Layers”, ICLR 2021
>
> So, to recap about impact: Any model and method that includes convolution or spectral analysis (in H) will be relevant to the current paper. Convolutions and Fourier are part of learning systems and representations that range from probabilistic, hierarchical models to deep neural nets. Hence, we believe that the “range of impact” of our work is extensive.
>
> On another note, the proof-of-concept application itself is useful in its own right – it can be seen as a plug-and-play improvement over practically *any* quaternionic CNN .

---

> ### Author Response · Authors · 2023-11-15
>
> *“First, the paper outlines the major difficulties of quaternions. However, I don't see why non-commutativity, two-sided convolution, versions of a convolution theorem, definition of determinant and the fact that $\mu = -1$ has infinite solutions are problems.”*
>
> Thank you for the question. These are problems because they “stand in the way” of generalizing results that are well-known in the real or complex domain to the domain of quaternions.
> Let me take eigenvalues and eigenvectors as an example. In C (set of complex numbers) we use the formula $Ax = λx$
> to define eigenvalues and eigenvectors. However in H (set of quaternions), we must distinguish between $Ax = λx$ and $Ax = xλ$ ; as multiplication is non-commutative, λx is not the same as xλ. It turns out that the two problems have completely different solutions in general (see for example Zhang 1997). Solutions for the one equation are termed left eigenvalues, and solutions for the other one are termed right eigenvalues. Furthermore, we do not have any a priori guarantee about the *number* of eigenvalues of each. (This is related to $\mu^2 = -1$ having infinite solutions).
>
> In the paper, we write that the convolution theorem can be written (and proved) easily in matrix form. However, this is not possible in H (Corollary 3.6.2). In particular,  “Writing proposition 3.5 in a matrix diagonalization form ($A = SΛS^{-1}$), where the columns of S are eigenvectors and Λ is a diagonal matrix of left eigenvalues, is not possible. This would require us to be able to write $CQ = Q Λ$. however, the right side of this equation computes right eigenvalues, while proposition 3.5 concerns left eigenvalues.” So, this is an example that looks “easy” in R or C, but becomes non-trivial and more nuanced / complicated in H.
>
> Concerning determinants, these can be a problem e.g. whenever we need to compute a normalizing constant for a probability distribution. In Normalizing flows (not studied in the paper), we need to define flows that let us compute a related determinant very fast. If in quaternions we are not even sure *how to define the determinant* (that is not to say that there is no related work on this, for example see [H. Aslaksen, "Quaternionic Determinants", 1999] or [Freeman J. Dyson, "Quaternion Determinants", 1972], then any possible generalization of NFs into H requires more work, definitely more than what would be perceived as “incremental”.

---

> ### Author Response · Authors · 2023-11-15
>
> *“If these issues are to be avoided, then why not go back to the real domain? Isn't the desired structure of quaternions what precisely results in these difficulties?”*
>
>
> This is correct. The structure of the quaternion algebra is the source of these difficulties.
>
> However, we cannot just go back to the real domain without abandoning the advantages of quaternions. In the context of NNs for example, probably the mainly cited advantage is the resulting compact, light-weight networks. (See for example Parcollet et al., “A survey of quaternion neural networks”, AI Review 2020).
>
> If what you mean by “going back” is assuming an isomorphism between H and R^4, and recasting everything as 4-dimensional vectors, again this won’t work without a cost. This is related to your other question:
>
> *“Lastly, what do the authors mean by [treating] signals in a holistic manner?”*
>
> By “holistic manner”, we mean that a multi-dimensional input may be treated as a single entity. For example, assuming that we have an image containing multiple channels, we need to write it as a structure of the form M x N x C . Strictly speaking, this is a tensor . Quaternions (and hypercomplex algebra in general) allow us to write this as a matrix.
> This may sound like a superficial difference, but it does bear advantages: The field of matrix methods is much more mature than tensor methods. Convolutions and Fourier in the form of matrices (Circulant, Fourier matrices) is one such example, and what this paper does is showing how a range of well-known related results generalize for quaternions. If we chose to stick with the tensor representation, we’d certainly be much more limited in scope. In Section 4, we discuss singular values of quaternion convolutional layers – this would be (currently?) impossible to do the exact same thing with tensors (there is no SVD for 3-way tensors). Cf. the excellent references on this topic:
> * G. Strang, “Linear Algebra and Learning from data”, 2019 (esp. chapter I.12)
> * C.Hillar and LH. Lim, “Most tensor problems are NP-hard”, J.ACM, 2013

---

> ### Author Response · Authors · 2023-11-15
>
> *“I acknowledge the representation power of quaternions, however, this paper does not exploit this representation in the context of learning. I feel that this work lacks a clearer connection between the stated contributions and the learning task [..]”*
>
> Allow us to respectfully disagree. In Section 4, we use additional, novel results (proofs in Appendix F), to improve learning of a quaternion-valued neural network. Our results show that there is significant improvement against not using our method. Specifically, we ran tests on CIFAR, where we show that our network, regardless of the hyperparameters, always obtains a boost in test performance. *Note that bounding the Lipschitz constant of convolutional QNNs is practically impossible without our theoretical results*. We discuss this further in the same section.
>
> *[..] possibly by showing examples where this representation makes a difference wrt standard (non-quaternion) methods.“*
>
> We agree that comparisons between quaternion and non-quaternion methods are in general interesting and useful. In this paper however, we do not try to show that quaternionic methods have this or that advantage; perhaps the discussion in Section 2, where we do touch upon pros and cons of quaternion algebra and its relation to learning systems, has been misleading in this respect. We could move it for example to a “less prominent” position in the paper, or make it an appendix.
>
> Concerning the debate between quaternions vs non-quaternions, this is the subject of numerous papers (see previous related answer on this thread, where we cite a number of works that do that). For example, we could have shown that a QNN can achieve similar results to a real-valued NN with only a fraction of its size. This is however already established in the literature.
>
> What we aim and *do* achieve in this paper, is proving a series of powerful theorems connecting convolution and Fourier analysis in H. This is done in Section 3. In Section 4, our results are “showcased” using an application in learning. In this context, we show that a QNN (known to have specific advantages against real-valued NNs) benefits from Lipschitz constant bounding, which is enabled by our theoretical results.

---

> ### Author Response · Authors · 2023-11-15
>
> *“I also acknowledge the application in Sec 4, however, the body of results presented in Sec 3 is far more general than that particular application.”*
>
> We do agree that the body for results in Section 3 is far more general in scope than the application of Section 4. Section 4 serves a “proof-of-concept” application, even though we do believe that it is important in itself. Let us note that this proof-of-concept however covers already 3 pages in the main paper + 4 pages in the appendix, in a paper of totally 27 pages – it only serves as one good example of how our results are useful and relevant to the learning representations community.

---

### Author Response · Authors · 2023-11-22
**Synopsis of rebuttal -- A."Is this paper relevant to the ICLR community?"**

We’d like to thank all the reviewers for their time, effort and useful suggestions. In this thread, we group the main points of rebuttal, in the form of summarizing questions and answers.
```
Question A: Is this paper relevant to the ICLR community ?
```
Answer: **Definitely yes. We believe that the paper is very relevant to the ICLR community**.

**First of all, this work is about two concepts – Fourier / spectral analysis and Convolution – that are at the very heart of a range of fields**, from signal processing to pattern recognition and learning systems and representations. Quaternion convolution in particular, is an operation that is prevalent in the vast majority of works on Quaternion Neural Networks (let me reference here the excellent survey of Parcollet, Morchid and Linarès, “A survey of quaternion neural networks”, AI Review 2020). The close relation between Fourier and Convolution has long been known for the real domain, and has been *extensively* used. In a nutshell, this paper shows how this relation generalizes to the quaternionic domain, and presents a proof-of-concept application of our theoretical results.

**Any model and method that includes convolution or spectral analysis (in H) will be relevant to the current paper**. Convolutions and Fourier are part of learning systems and representations that range from probabilistic, hierarchical models to deep neural nets. Hence, we believe that the “range of impact” of our work is extensive.

We have taken a conscious decision to include this specific application – it combines a host of themes, like **Quaternions in NNs, bounding the Lipschitz constant, and analyzing singular values of convolutional layers. The latter two have been the subject of interest for the ICLR / NIPS / ICML etc. community**. For example:

* Sedghi, Gupta, Long, “The singular values of convolutional layers”, ICLR 2019

* Singla, Feizi, “Fantastic Four: Differentiable and Efficient Bounds on Singular Values of Convolution Layers”, ICLR 2021

On another note, the proof-of-concept application itself is useful in its own right – it can be seen as a plug-and-play improvement over practically any quaternionic CNN .

---

> ### Author Response · Authors · 2023-11-22
> **Synopsis of rebuttal - B. "Aren't the theoretical results incremental?"**
>
> ```
> Question B: Aren’t the theoretical results incremental?
> ```
>
> In a nutshell, **this paper is the first work to explore the eigenstructure of quaternionic Convolution and Fourier matrices. We show that their relation is much more complex with respect to the well-known theorems in R and C.**
> In defense of our theoretical results, we bring up a number of examples, which we believe are characteristic:
>
> * Proposition 3.5 is about the **eigenstructure of Quaternionic Circulant matrices**, and their connection to the QFT matrix. The connection between Circulants and the DFT matrix is well-known for R or C – in H (set of quaternions), eigenstructure is however necessarily a far more nuanced issue. For a start, we have two different spectra: a left spectrum and a right spectrum, corresponding to left and right eigenvalues respectively. *This is a feature unique to H*. The two spectra are in general different from the other, and no connection is known to hold between them, in terms of, say, computing the left spectrum if one knows about the right spectrum, or vice versa. The literature has mostly explored the right spectrum, for which the have been proposed numerical computation algorithms. For the left spectrum however, works are far more scarce. Proposition 3.5 tells us that Quaternionic Circulant matrices and the QFT are related through the left spectrum of the former, however there has been no hitherto algorithm to compute left eigenvalues (apart from toy examples of 2x2 or 3x3 matrices). *In quaternion matrix algebra, a field which dates as far back as the first half of the $20^{th}$ century, to our knowledge this is the first work where a computational scheme to compute the left spectrum is proposed* (aside the noted small-matrix cases, cf. So, “Quaternionic Left Eigenvalue Problem”, 2005; Marcias-Virgos, “Rayleigh quotient and left eigenvalues of quaternionic matrices”, Linear & Multilinear Algebra 2023). Furthermore, **it is shown to be applied to a real, relevant learning problem**.
>
> * In the paper, we write that the convolution theorem for R or C can be written (and proved) easily in matrix form. However, this is not possible in H (Corollary 3.6.2). In particular,  “Writing proposition 3.5 in a matrix diagonalization form (A = SΛS−1), where the columns of S are eigenvectors and Λ is a diagonal matrix of left eigenvalues, is not possible. This would require us to be able to write $CQ = QΛ$; however, the right side of this equation computes right eigenvalues, while proposition 3.5 concerns left eigenvalues.” **So, this is another example that looks “easy” in R or C, but becomes non-trivial and more nuanced / complicated in H**.
>
> * In Proposition 3.7iv, we state (and prove) that if $\lambda$ and $\kappa$ are eigenvalues of circulant matrices L and K, then their *product* LK has the eigenvalue: $\lambda^\nu \kappa^\mu$, where $\nu = \kappa \mu \kappa^{-1}$. This is a result that has no analogue in R or C, and only makes sense in H. Furthermore it generalizes smoothly into H: In a commutative algebra we would have $\kappa$ and $\kappa^{-1}$ cancel out, which doesn’t happen for quaternions. We believe that this also a non-obvious result a priori: A first guess could be that $\lambda^\mu \kappa^\mu$ is the eigenvalue, which turns out to be wrong.
> In proposition 4.3 and corollary 4.4.1, the matrix $\Xi$ is introduced, in order to enable computation and bounding of the required spectra. Note the form:    $\begin{bmatrix}
>    	 \lambda_m^{\parallel} & \lambda_n^{\perp} \\\\
>    	 \lambda_m^{\perp}     & \lambda_n^{\parallel} \\\\
>     \end{bmatrix}$.
>
> **There is no analogue in R or C, as simplex and perplex components make sense only in H. Furthermore, note that this elegantly generalizes from R/C**: In these domains, the off-diagonal (perplex) components would be zero, and computation would be trivial.

---

> ### Author Response · Authors · 2023-11-22
> **Synopsis of the rebuttal - C. "How are the theoretical results connected to learning applications? There is a lack of convincing applications."**
>
> ```
> Question C: "How are the theoretical results connected to learning applications? There is a lack of convincing applications."
> ```
>
> **We chose this specific application for two reasons: it was optimal in terms of being able to integrate the vast majority of our novel theoretical results;  furthermore, it is about an application that combines topics of interest to different subsets of the learning community**: Computing and bounding a Lipschitz constant; singular values of network layers; quaternionic convolutional neural networks. The proposed application would be impossible without our novel results. Please also see point B regarding the significance of our contributions.
>
> * Propositions 3.1 and 3.2 are about left-side quaternion convolution being expressed as a circulant matrix. The bulk of the NN layers uses left-side quaternion convolution. These will be treated as products of the form $Cx$, where $C$ is circulant, and use the subsequent results for circulant matrices.
>
> * Proposition 3.3 is about the matrix form of the QFT, and 3.4 is about a geometric intuition over this form.  Proposition 3.3 is used in conjunction with..
>
> * ..Proposition 3.5, where *left eigenvalues* of C are shown to be computable via a right QFT. The corresponding eigenvectors are vectors of the QFT matrix we discussed in 3.3, 3.4. **This computation is at the heart of the application in Section 4 - none of the computations would be possible without it**.
>
> * Propositions 3.8 and 3.9 are about doubly-block circulant matrices – again, this is absolutely a requirement as we are dealing with 2D inputs (CIFAR images) in the experiments.
>
> * All the theoretical results in Section 4 are there specifically to motivate and enable the Lipschitz bounding application. We show how we can clip singular values of quaternionic convolution, by the use of an auxiliary “Ξ” matrix.
>
> **The proof-of-concept application of Section 4 is itself useful in its own right – it can be seen as a plug-and-play improvement over practically any quaternionic CNN**. Our results show that the proposed technique leads to more efficient training and network accuracy at negligible overhead.

---

> ### Author Response · Authors · 2023-11-22
> **Synopsis of the rebuttal - D. "Why should a quaternion-oriented work interest us in the first place?"**
>
> ```
> Question D -- "Why should a quaternion-oriented work interest us in the first place?"
> ```
>
> Let us clarify that in Section 2 we merely provide a short discussion on the pros and cons of using quaternions as part of a learning representation framework. Perhaps it has been misunderstood that the point of the paper is arguing that “quaternions are worth using”. We have included Section 2 as a short intro for scholars that have perhaps heard that quaternions are useful for e.g. representing rotations, but do not know about the *modern* uses of quaternions in learning systems.
>
> **The paper is not about whether quaternions are useful or not**; we believe that this has been extensively argued in previous literature. There is a host of works on ICLR and relevant venues with their main focus on quaternions:
>
> * Parcollet, Ravanelli, Morchid, Linarès, Trabelsi, De Mori and Yoshua Bengio: “Quaternion Recurrent Neural Networks”, ICLR 2019
>
> * Zhang, Tay, Zhang, Chan, Luu, Hui and Fu: “Beyond fully-connected layers with quaternions: Parameterization of hypercomplex multiplications with 1/n parameters”, ICLR 2021
>
> * Zhang, Tao, Yao, Liu, “Quaternion knowledge graph embeddings”, NeurIPS 2019
>
> * Miron, Flamant, Le Bihan, Chainais and Brie, “Quaternions in Signal and Image Processing”, IEEE Signal Processing Magazine 2023
>
> * Qin, Zhang, Xu, Xu, “Fast Quaternion Product Units for Learning Disentangled Representations in SO3”, IEEE Transactions on Pattern Analysis and Machine Intelligence, 2023
>
> The first and second works have been published in ICLR. The second one has got an “outstanding paper award” in ICLR. Hence, quaternions in the context of learning representations have indeed interested the community, and have been very much welcomed by it in the recent past.
>
> In the context of representing high-dimensional signals (e.g. neural network inputs or intermediate layers or color images) via elements of H, the main advantage is that we are allowed to treat them in a “holistic manner”.
>
> By “holistic manner”, we mean that a multi-dimensional input may be treated as a single entity. For example, assuming that we have an image containing multiple channels, we need to write it as a structure of the form M x N x C . Strictly speaking, this is a *tensor*. Quaternions (and hypercomplex algebra in general) allow us to write this as a matrix.
>
> This may sound like a superficial difference, but it does bear advantages: The field of matrix methods is much more mature than tensor methods. Convolutions and Fourier in the form of matrices (Circulant, Fourier matrices) is one such example, and what this paper does is showing how a range of well-known related results generalize for quaternions. If we chose to stick with the tensor representation, we’d certainly be much more limited in scope. In Section 4, we discuss singular values of quaternion convolutional layers – this would be (currently?) impossible to do the exact same thing with tensors (there is no SVD for 3-way tensors). Cf. the excellent references on this topic:
>
> * G. Strang, “Linear Algebra and Learning from data”, 2019 (esp. chapter I.12)
>
> * C.Hillar and LH. Lim, “Most tensor problems are NP-hard”, J.ACM, 2013

---

### Meta-Review · Area_Chair_J8VQ · 2023-12-06

**Metareview:**

The paper presents some mathematical results on the connections between, for example, the Fourier Transform matrix with quaternion entries and the standard Discrete Fourier transform matrix.

Strengths:
 - The novelty of these observations and proofs is not disputed by the reviewers

Weaknesses:
 - The importance of the results is not considered by the reviewers of sufficient magnitude for dissemination at ICLR.

As noted by the authors, ICLR is indeed a venue where theoretical results are welcome.  However, what is key is the magnitude of such results, and the breadth of their application.  To take Sedghi et al 2019 as an example, that paper applies to a much more widely used network architecture, with a practical computation scheme, and its CIFAR experiments apply to architectures much closer to the state of the art.  That the current paper can apply only to a highly restricted variant of ResNet is a further indication that its results apply only to a small subset of architectures of relevance to the ICLR community.

In summary, I see no inconsistency in the three reviews.  The reviewers have engaged with the paper, actively seeking to understand its relevance.   Not all reviewers may have consumed every detail of the 27 pages, but their objections are all to do with the relevance of the results, not with the detail.

**Justification For Why Not Higher Score:**

The paper needs significant rework in order to clearly show why its derivations have practical benefit.

**Justification For Why Not Lower Score:**

n/a

---

### Decision · Program_Chairs · 2024-01-16

Reject